# Ligand-coupled conformational changes in a cyclic nucleotide-gated ion channel revealed by time-resolved transition metal ion FRET

Pierce Eggan, Sharona E Gordon*, William N Zagotta*

Department of Physiology and Biophysics, University of Washington, Seattle, United States

*For correspondence:
seg@uw.edu (SEG);
zagotta@uw.edu (WNZ)

Competing interest: The authors declare that no competing interests exist.

## eLife Assessment

This **valuable** study uses fluorescence lifetime imaging and steady-state and time-resolved transition metal ion FRET to characterize conformational transitions in the isolated cyclic nucleotide binding domain of a bacterial CNG channel. The data are **compelling** and support the authors' conclusions. The results advance the understanding of allosteric mechanisms in CNBD channels and have theoretical and practical implications for other studies of protein allostery. A limitation is that only the cytosolic fragments of the channel were studied.

**Abstract** Cyclic nucleotide-binding domain (CNBD) ion channels play crucial roles in cellular-signaling and excitability and are regulated by the direct binding of cyclic adenosine- or guanosine-monophosphate (cAMP, cGMP). However, the precise allosteric mechanism governing channel activation upon ligand binding, particularly the energetic changes within domains, remains poorly understood. The prokaryotic CNBD channel SthK offers a valuable model for investigating this allosteric mechanism. In this study, we investigated the conformational dynamics and energetics of the SthK C-terminal region using a combination of steady-state and time-resolved transition metal ion Förster resonance energy transfer (tmFRET) experiments. We engineered donor-acceptor pairs at specific sites within a SthK C-terminal fragment by incorporating a fluorescent noncanonical amino acid donor and metal ion acceptors. Measuring tmFRET with fluorescence lifetimes, we determined intramolecular distance distributions in the absence and presence of cAMP or cGMP. The probability distributions between conformational states without and with ligand were used to calculate the changes in free energy ($\Delta G$) and differences in free energy change ($\Delta\Delta G$) in the context of a simple four-state model. Our findings reveal that cAMP binding produces large structural changes, with a very favorable $\Delta\Delta G$. In contrast to cAMP, cGMP behaved as a partial agonist and only weakly promoted the active state. Furthermore, we assessed the impact of protein oligomerization and ionic strength on the structure and energetics of the conformational states. This study demonstrates the effectiveness of time-resolved tmFRET in determining the conformational states and the ligand-dependent energetics of the SthK C-terminal region.

## Introduction

CNBD channels, part of the voltage-gated ion channel superfamily, play pivotal roles in sensory perception, signal transduction, and cellular excitability (*Craven and Zagotta, 2006*). The CNBD family includes cyclic nucleotide-gated (CNG) channels, which are responsible for visual and olfactory

signal transduction, and hyperpolarization-activated cyclic nucleotide-gated (HCN) channels, which regulate the pacemaker activity of the heart (*Matulef and Zagotta, 2003*; *He et al., 2014*; *Kaupp and Seifert, 2002*). Despite their diverse physiological roles, CNBD channels all possess a shared tetrameric structure with four identical or similar subunits surrounding a central ion-conducting pore. Each subunit contains three regions: an N-terminal region, a transmembrane region with a voltage-sensor and pore domain, and a cytosolic C-terminal region that consist of a CNBD plus a C-linker connecting the CNBD to the pore (*James and Zagotta, 2018*). These channels are activated by the binding of cyclic nucleotides (such as cAMP or cGMP) to the CNBD, which induces conformational changes throughout the protein structure to ultimately open the pore. This ligand-induced allosteric opening is still poorly understood, and questions remain about the structural and energetic changes that occur during this allosteric regulation. In particular, what are the energetics of the conformational changes in individual domains, how are these conformational changes coupled within and between subunits, and how do these processes differ for full and partial agonists?

SthK is a prokaryotic member of the CNBD family from *Spirochaeta thermophila,* which has considerable sequence and structural similarity to eukaryotic CNBD channels and offers a powerful model for better understanding the allosteric regulation in these channels (*Brams et al., 2014*). It is easily expressed in *E. coli*, has established biochemical purification methods, and its physiological properties have been studied with ion flux assays and patch-clamp electrophysiology (*Morgan et al., 2019*; *Schmidpeter et al., 2018*). We have also characterized a cysteine-free version of the protein (cfSthK) and found it to behave nearly identically to wild-type channels, making it amenable to thiol-based site-specific labeling (*Morgan et al., 2019*). As previously shown, binding of cAMP to SthK causes robust channel currents in *E. coli* spheroplasts (*Figure 1A*) with a high degree of cooperativity (*Figure 1B*; Hill slope, *h*: 2.9±0.2; *Morgan et al., 2019*). In contrast, cGMP appears to be a poor partial agonist for SthK, and its binding to the same structural pocket is only weakly coupled to conformational changes that increase channel open probability (*Figure 1B*).

X-ray crystallography and cryo-EM structures of SthK reveal notable ligand-induced structural changes throughout the protein, particularly in the CNBD (*Kesters et al., 2015*; *Rheinberger et al., 2018*; *Gao et al., 2022*). These differences include a rotation of the CNBD C-helix towards the β-roll upon binding of cyclic nucleotide. This important conformational change is thought to be coupled to a rearrangement of the C-linker, and ultimately to the opening of the pore. However, structures alone have not been able to fully characterize the energetic landscape of the allosteric regulation, especially in determining the structural heterogeneity within conformational states and the energetics between states.

To link structural information to the mechanism of allostery, we used a simple four-state model to describe the conformational states and energetics in the C-terminal of SthK (*Figure 1C*). This framework has been used previously to describe allosteric proteins generally and the ligand-binding domains of CNBD channels in particular (*DeBerg et al., 2016*; *Collauto et al., 2017*; *Colquhoun, 1998*; *Auerbach, 2003*; *Aviram et al., 2018*; *Guo and Zhou, 2016*). In this model, the CNBD can exist in either resting or active conformational states, both of which can be ligand free (apo) or ligand bound (holo). Whereas in the absence of ligand the transition from resting to active is unfavorable, the presence of agonist (either full or partial) makes the active state more favorable. The equilibrium constants, and therefore energetics, that describe the transitions between states determine the steady-state fraction of molecules in each given state. Using the four-state energetic model as a framework and an experimental method for determining the fraction of molecules in each state, we can calculate the changes in free energy (ΔG) and differences in energy change (ΔΔG) between the conformational states of the CNBD upon binding of ligands like cAMP and cGMP.

A complex protein, such as an ion channel, consists of multiple domains, each of which can undergo conformational changes that are coupled to ligand binding, transmembrane voltage, or the state of a neighboring domain (*Horrigan and Aldrich, 2002*; *Hofmann, 2023*). For example, allostery in CNBD channels can be modeled as a cyclic-nucleotide-dependent conformational change in each subunit CNBD that is coupled to a concerted conformational change in the C-linker, which in turn is coupled to a opening conformational change in the pore (*Craven and Zagotta, 2006*; *DeBerg et al., 2016*). In this context, to truly understand the allosteric mechanism in CNBD channels, we would need to know ΔG and ΔΔG for each of these domains. Therefore, we need a method that can measure the conformational energetics in each protein domain.

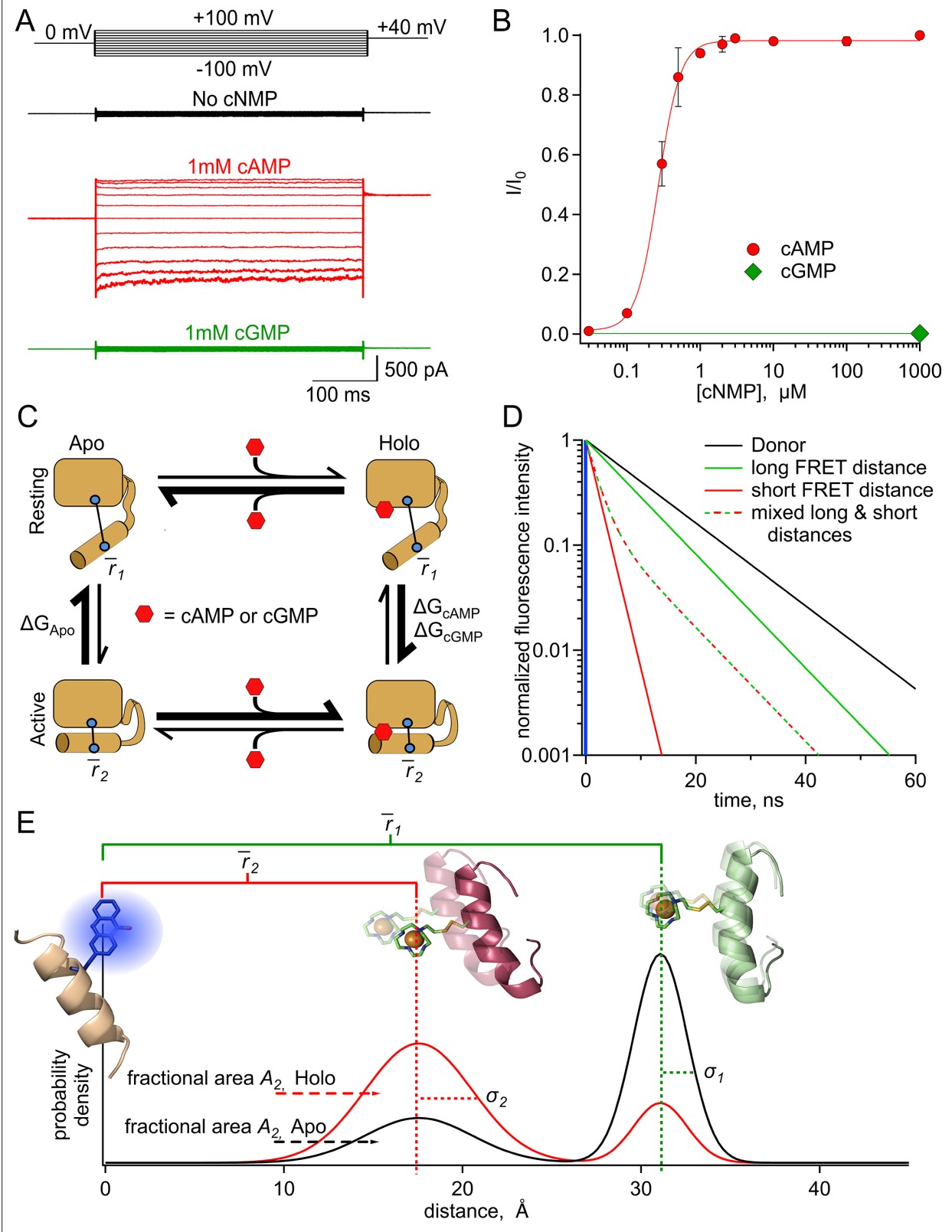

**Figure 1.** SthK as a model protein for characterizing the allosteric regulation in CNBD channels. (**A**) Representative macroscopic A208V-cfSthK currents in inside-out patches from bacterial spheroplasts in response to voltage steps shown at the top, in the absence of cyclic nucleotide (top black traces), in saturating 1 mM cAMP (middle red traces), and in saturating 1 mM cGMP (bottom green traces). (**B**) Dose-response relation of A208V-cfSthK for cAMP at +80 mV (red circles, n=6), fit with the Hill Equation (red curve, $K_{1/2}$: 0.27±0.01 μM, slope: 2.9±0.2, means ± SD). Fractional activation by 1 mM cGMP

*Figure 1 continued*

was 0.002±0.003 (mean ± SD; green diamond, n=6). (**C**) Diagram showing four states (either apo or holo, and either resting or active) of the CNBD and the associated ΔG's for the transitions between states. (**D**) Theoretical fluorescence lifetime decays of a donor fluorophore in the time-domain showing basis of time-resolved tmFRET. A single exponential donor and mixtures of two single tmFRET distances (short and long) are shown. (**E**) Theoretical distance distributions showing two states with average distances, $\bar{r}$, heterogeneity within each conformational state as standard deviation, $\sigma$, and heterogeneity between conformational states as fractional area, $A_2$, for apo (black) and holo (red) conditions.

The online version of this article includes the following source data for figure 1:

**Source data 1.** Excel data for representative electrophysiology traces (***Figure 1A***) and summary of cyclic nucleotide dose response (***Figure 1B***).

In this study, we utilized our recently described steady-state and time-resolved tmFRET methods to measure the energetics in an isolated C-terminal fragment of SthK (***Zagotta et al., 2021***; ***Gordon et al., 2024***; ***Zagotta et al., 2024***). In tmFRET, a donor fluorophore is paired with a transition metal ion, such as $Cu^{2+}$, $Fe^{2+}$, or $Ru^{2+}$, as an acceptor. Donor-acceptor pairs can then be incorporated in a protein at different residue positions. As in classical FRET with two fluorophores, the efficiency of energy transferred between the donor fluorophore and acceptor metal ion is steeply distance dependent and can be used as a molecular ruler to measure distances between sites on a protein, and therefore conformational changes (***Lakowicz, 2006***).

In time-resolved tmFRET, changes in fluorescence lifetimes of the donor fluorophore are quantified to report FRET efficiencies, and therefore molecular distances between the donor and acceptor. The fluorescence lifetime is the time, in nanoseconds, between excitation of the donor fluorophore and emission of a photon (***Lakowicz, 2006***). Unlike steady-state FRET, time-resolved FRET can provide nanosecond snapshots of the distribution of FRET efficiencies in a sample (***Zagotta et al., 2021***; ***Zagotta et al., 2024***). In the time-domain, a histogram of photon latencies for the simplest lifetimes appears as a single exponential decay (***Figure 1D***, black line). FRET resulting from a single donor-acceptor distance also produces lifetimes described by a single exponential decay, but with a time constant that decreases in proportion to the FRET efficiency (***Figure 1D***, solid green and red lines). However, in the case of two donor-acceptor distances, such as from two protein conformations, the intensity decay will be double exponential, with contributions from both donor-acceptor distances. The average distance of each component and the fraction among components can be determined by fitting the decay with a double-exponential decay model. The fractional contribution of each lifetime represents the prevalence of each distance, and thus conformational state, in the sample population (***Figure 1D***, green-red dashed lifetime is a mix of 20% solid green and 80% solid red lifetimes; ***Lakowicz, 2006***). Time-resolved tmFRET, therefore, resolves the structural distances and relative abundance of multiple conformational states in a protein sample.

Under physiological conditions, donor-acceptor distances in proteins will typically be heterogeneous and fluorescence lifetime data will reflect this heterogeneity. Thus, we have fit lifetime data with a FRET model that assumes Gaussian distributions of donor-acceptor distances within each state. This heterogeneity can arise from both: (1) backbone and rotameric variations within a given state (the width of each peak, ***Figure 1E***) and (2) differences between conformational states (relative fraction of the resting, black, and active, red, peaks, ***Figure 1E***). Heterogeneity between conformational states reflects the proportion of molecules in the resting and active states of the protein, which can then be used to calculate the free energy difference between the states (ΔG). In this study, we used various donor-acceptor pairs in the CNBD of SthK to measure distance distributions of the C-helix relative to the β-roll in the absence and presence of ligand. Distance distributions were used to determine the energetics for the transitions between conformational states of the four-state model described above. These results provide a more complete understanding of the structural and energetic changes in the CNBD with ligand binding.

## Results

We engineered three donor sites for tmFRET experiments into a C-terminal fragment of SthK (SthK$_{Cterm}$), comprised of the C-linker and CNBD domains (***Kesters et al., 2015***). At these sites (359, 361, and 364), we introduced the unnatural amino acid acridon-2-ylalanine (Acd) as a donor fluorophore using amber codon suppression with a previously described tyrosyl tRNA synthetase (***Zagotta et al., 2024***; ***Sungwienwong et al., 2017***). Incorporation of Acd into stop-codon containing SthK$_{Cterm}$ constructs

occurred only in the presence of both the amino-acyl tRNA synthetase/tRNA plasmid (RS/tRNA) and the Acd amino acid. SDS-PAGE followed by in-gel fluorescence imaging (SthK$_{Cterm}$-S361-TAG) indicated that Acd was site-specifically incorporated into our SthK$_{Cterm}$ constructs and the incorporated product was readily purified for use in tmFRET experiments (*Figure 2A*).

We introduced metal ion acceptor binding sites for tmFRET using cysteine mutations in SthK$_{Cterm}$ for modification with cysteine-reactive metal chelators (TETAC and phenanthroline maleimide [phenM]) bound to different transition metals (Cu$^{2+}$, Fe$^{2+}$, and Ru$^{2+}$). The three different metal ion-acceptor complexes used in this study, [Cu(TETAC)]$^{2+}$, [Fe(phenM)$_3$]$^{2+}$, and [Ru(bpy)$_2$phenM]$^{2+}$, are shown in *Figure 2B*. The distance dependence of energy transfer between donor and acceptor is described by the Förster equation and is dependent on the overlap between the absorbance spectrum of the acceptor and the emission spectrum of the donor, among other factors (*Lakowicz, 2006*; *Stryer and Haugland, 1967*). Paired with Acd, these transition metal acceptor complexes resulted in distances with 50% energy transfer (R$_0$) of 15.6 Å, 41.8 Å, 43.5 Å for [Cu(TETAC)]$^{2+}$, [Fe(phenM)$_3$]$^{2+}$, and [Ru(bpy)$_2$phenM]$^{2+}$, respectively. Thus, with Acd as a donor, these acceptors allowed us to measure both short (10–20Å) and long (25–50Å) distances across SthK$_{Cterm}$ (*Figure 2B*; *Gordon et al., 2024*).

The SthK$_{Cterm}$ fragment tetramerizes in solution at higher protein concentrations, even though the transmembrane domains are absent. Although the tetrameric structure is the physiologically relevant state for the channel, tetramerization introduces challenges in minimizing the contribution of inter-subunit FRET measurements where only intra-subunit donor-acceptor pairs are desired. Using tmFRET, which is sensitive to relatively short distances, mitigates but does not eliminate this concern. To avoid potential contributions from inter-subunit FRET, Acd-labeled protein was tetramerized with an excess (at least a 3:1 molar ratio) of wild-type SthK$_{Cterm}$ protein (WT; *Figure 2C*). WT protein had neither Acd incorporation nor cysteine mutations. This approach ensured that, on average, no more than one Acd-cysteine-containing subunit was present in each tetramer, and our FRET efficiency measurements would be almost exclusively from donor-acceptor pairs within the same subunit.

Tetrameric SthK$_{Cterm}$ was efficiently separated from monomeric protein using size exclusion chromatography (SEC), where tetrameric protein eluted primarily as a single monodispersed peak at 14 mL, as observed by both absorbance at 280 nm and Acd fluorescence (*Figure 2D*, closed triangle). We confirmed this large SEC peak corresponded to tetramers of SthK$_{Cterm}$ using mass photometry, which reported a mass of ~105 kDa, compared to a predicted mass of 101 kDa (*Figure 2E*). At the low concentrations (~10 nM) used for mass photometry, a second small peak was observed of ~30 kDa, which is below the analytical range for this method. All tmFRET experiments used higher protein concentrations to ensure tetramerization.

## Steady-state tmFRET for determining weighted-average distance changes

To resolve the structural changes in tetrameric SthK$_{Cterm}$, we characterized steady-state tmFRET using SthK$_{Cterm}$-S361Acd-V416C, with a long-distance donor-acceptor pair (*Figure 3A*), and SthK$_{Cterm}$-Q364Acd-R417C, with a shorter distance pair (*Figure 3C*). We used the metal ion acceptors [Fe(phenM)$_3$]$^{2+}$ and [Ru(bpy)$_2$phenM]$^{2+}$ for the long-distance pair and [Cu(TETAC)]$^{2+}$ for the short-distance pair. We measured time courses of Acd fluorescence intensity of the cysteine-containing construct, upon addition of the acceptor, relative to protein without the cysteine mutation, measured separately as a negative control. We plotted the ratio of the normalized fluorescence intensity for the two protein samples as F$_{Cys}$/F$_{No Cys}$. The addition of metal acceptor produced pronounced quenching specifically in the cysteine-containing constructs (*Figure 3A, B and C*). This fluorescence decrease is indicative of FRET-based quenching of the donor by the metal acceptor, with the FRET efficiency given by E=1- F$_{Cys}$/F$_{No Cys}$. The addition of a saturating concentration of cAMP (160 μM) further increased quenching, indicating a greater FRET efficiency in the presence of agonist compared to the apo state. The increased FRET is consistent with a cAMP-induced decrease in donor-acceptor distance as the C-helix moves towards the β-roll. The dependence of the apparent FRET efficiency change on the cAMP concentration was measured for each donor-acceptor pair, then normalized for comparison and fit with the Hill equation (*Figure 3D*). The Hill fits obtained from different sites and acceptor metals are in close agreement with one another, suggesting similar binding affinities and that our previous measurements were obtained at saturating concentrations. Additionally, the three Hill fits with slopes, *h*, of ~1 suggest that the cooperativity observed in full-length channel electrophysiology

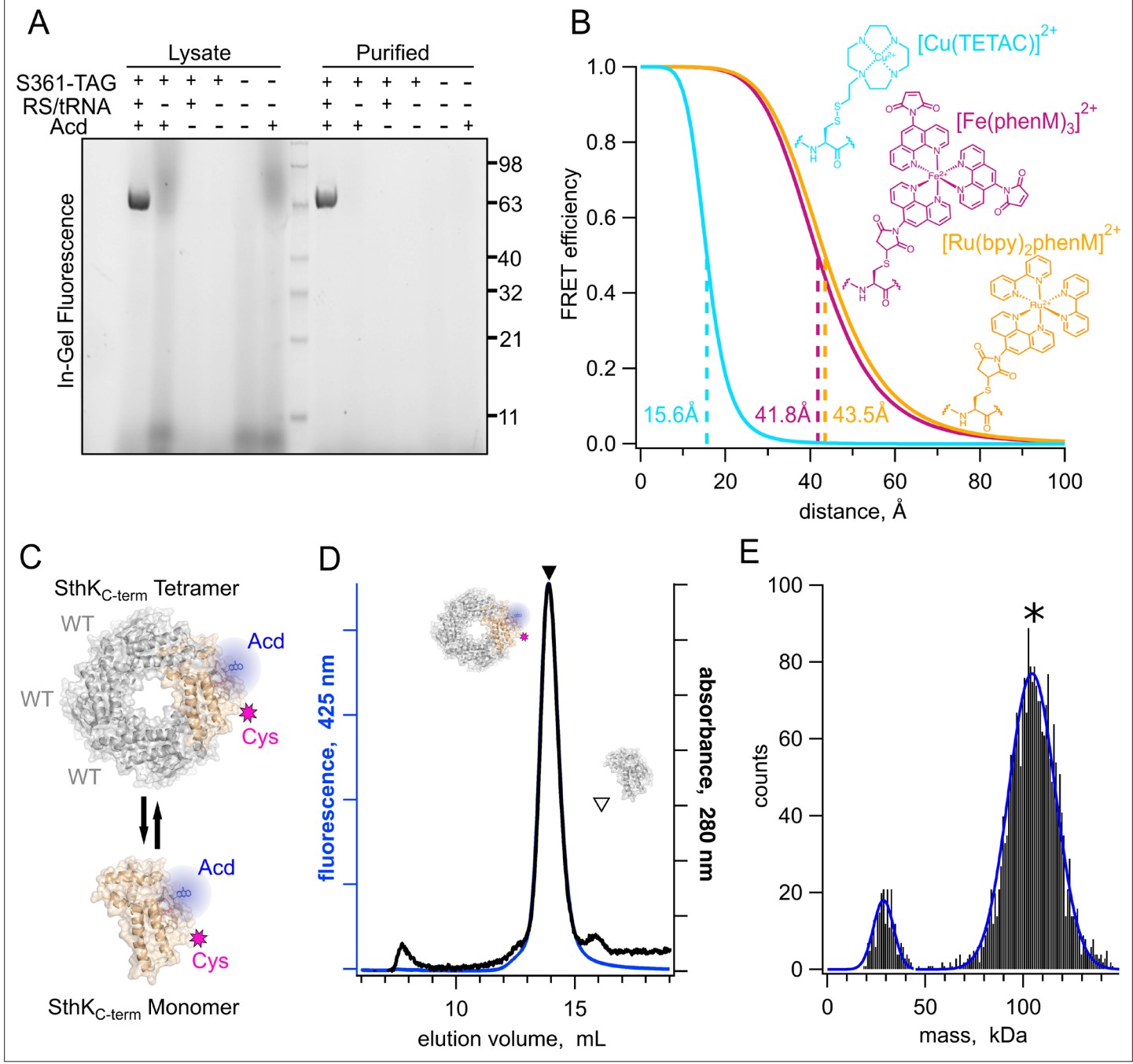

**Figure 2.** Expression, purification, and analysis of tetrameric SthK_Cterm. (**A**) In-gel protein fluorescence showing selective Acd incorporation into SthK_Cterm in the absence and presence of S361Acd TAG site, Acd aminoacyl tRNA synthetase/tRNA (RS/tRNA), and unnatural amino acid Acd. (**B**) Structures of cysteine modified by acceptor compounds [Cu(TETAC)]²⁺ (cyan), [Fe(phenM)₃]²⁺ (magenta), and [Ru(bpy)₂phenM]²⁺ (orange) along with their corresponding Förster curves of FRET efficiency as a function of distance from Acd, and their R₀ values specified and marked with dashed lines. (**C**) SthK_Cterm cartoon as tetramer and monomer showing WT subunits in gray and cysteine-containing Acd-labeled subunits in tan. (**D**) SEC traces (absorbance at 280 nm in black and 425 nm fluorescence emission for Acd in blue) of isolated WT-Acd-heterotetrameric protein (closed triangle) vs monomeric WT protein (open triangle). (**E**) Mass photometry histogram data showing primarily tetramers (*), with single Gaussian fits (blue, 29±7.3 kDa and 104.5±16.8 kDa , means ± SD).

The online version of this article includes the following source data for figure 2:

**Source data 1.** Original uncropped protein gel image (*Figure 2A*).

**Source data 2.** Labeled cropped protein gel image (*Figure 2A*).

**Source data 3.** Excel data for size exclusion chromatography (*Figure 2D*) and mass photometry data (*Figure 2E*).

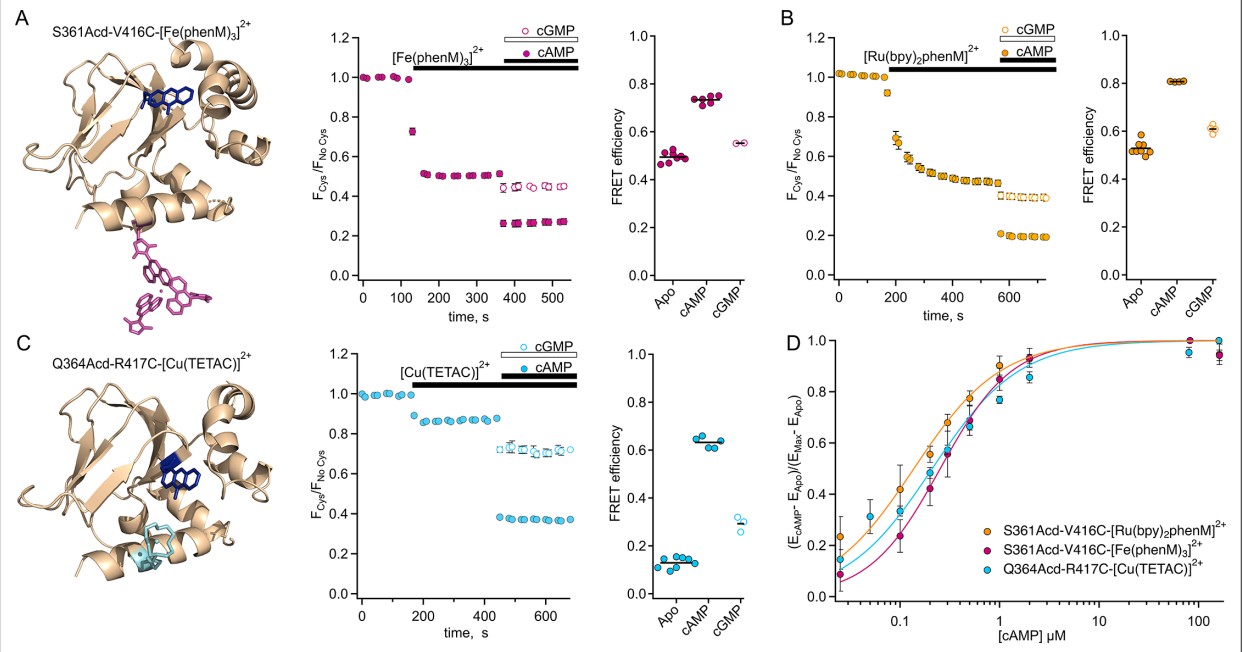

**Figure 3.** Steady-state tmFRET data from tetrameric SthK$_{Cterm}$. (**A**) Left: structure of one subunit of SthK$_{Cterm}$-S361Acd with [Fe(phenM)$_3$]$^{2+}$ acceptor incorporated at V416C (adapted from PDB: 4D7T; *Kesters et al., 2015*). Middle: Time course of average fluorescence upon addition of [Fe(phenM)$_3$]$^{2+}$ and then cAMP or cGMP (apo, n=8; cAMP n=6; and cGMP, n=2). Right: summary of the FRET efficiencies from individual experiments, with mean values as horizontal lines. (**B**) Left: Time course of average fluorescence for same site upon addition of [Ru(bpy)$_2$phenM]$^{2+}$ acceptor and then cAMP or cGMP (apo, n=8; cAMP, n=4; and cGMP, n=4). Right: summary of FRET efficiencies, with mean values as horizontal lines. (**C**) Left: structure of one subunit of SthK$_{Cterm}$-Q364Acd-417C with [Cu(TETAC)]$^{2+}$ acceptor incorporated at R417C. Middle: Time course of averaged fluorescence upon addition of [Cu(TETAC)]$^{2+}$ then cAMP or cGMP (apo, n=8; cAMP, n=5; and cAMP, n=3). Right: summary of FRET efficiencies, with mean values as horizontal lines. (**D**) Dose response relations of FRET efficiency change as a function of cAMP concentration normalized for comparison and fit with Hill equations ($K_{1/2}$: 0.25±0.01 μM, 0.14±0.01 μM, 0.21±0.02 μM, and $h$:1.2±0.07, 1, and 1 for [Fe(phenM)$_3$]$^{2+}$, magenta; [Ru(bpy)$_2$phenM]$^{2+}$, orange; and [Cu(TETAC)]$^{2+}$, cyan respectively, ± SD).

The online version of this article includes the following source data and figure supplement(s) for figure 3:

**Source data 1.** Excel data for time courses and dot plots of steady-state FRET efficiencies and dose response (*Figure 3A–D*).

**Figure supplement 1.** Steady-state tmFRET of monomeric and tetrameric SthK$_{Cterm}$.

**Figure supplement 1—source data 1.** Excel data for size exclusion chromatography traces, monomer and tetramer steady-state time courses, and FRET efficiency summaries (*Figure 3—figure supplement 1A–D*).

is not observed in the isolated C-terminal fragment. These steady-state tmFRET results across different donor-acceptor sites and acceptor complexes underscore the large conformational change of the C-helix induced by cAMP.

In addition to measuring the conformational rearrangement with the full agonist cAMP, we also used the partial agonist cGMP. Like cAMP, cGMP increased quenching by our transition metal labels (i.e. increased FRET); however, the increase was much smaller than that produced by cAMP. As discussed above, cGMP is a poor partial agonist of SthK which elicits only a small fraction of the channel current elicited by cAMP (*Figure 1A*). The cGMP-induced increase in FRET we observed provides further evidence that cGMP indeed binds to SthK and that cGMP binding causes a conformational change that moves the C-helix closer to the β-roll. These data do not, however, distinguish between (1) cGMP producing a smaller movement relative to cAMP (a distinct state), and (2) cGMP producing the same size movement but with a lower fraction in the active state (different energetics).

At lower protein concentrations, monomeric SthK$_{C-term}$ protein can be separated from tetrameric protein on SEC. Interestingly, for multiple donor-acceptor sites, steady-state tmFRET measurements consistently revealed higher FRET efficiencies (shorter distances) for monomers than for tetramers, particularly in the apo state (*Figure 3—figure supplement 1*). These experiments suggest that tetramerization destabilizes the resting-to-active transition of the SthK$_{C-term}$ protein. For our remaining

experiments, we focused on the conformational changes in the tetrameric state of the protein, which is more physiologically relevant.

## Predicted in silico distance distributions

To validate the accuracy of tmFRET in determining donor-acceptor distances and ligand-dependent changes in distances, we compared our data to predictions based on structural models. For each donor-acceptor pair, we modelled the donor and acceptor labels on the resting and active static structures of SthK$_{Cterm}$ (PDB: 7RSH and 4D7T, respectively) to obtain in-silico weighted rotameric ensembles utilizing a previously described python package, chiLife (*Figure 4A*; *Kesters et al., 2015*; *Gao et al., 2022*; *Tessmer and Stoll, 2023*). We then calculated the distance distributions between every Acd rotamer and every acceptor rotamer for both the resting and the active structures within subunits (intra-subunit) and between all other subunits (inter-subunit; *Figure 4B*). The inter-subunit distances predicted by chiLife for these sites were longer than 50 Å, contributing less than 15% FRET efficiency to the intra-subunit FRET in either the resting or active states. This inter-subunit FRET is predicted to be negligible in our experiments with added WT SthK$_{Cterm}$.

We compared the intra-subunit chiLife distributions with our measured weighted-average distances predicted from steady-state tmFRET experiments. We calculated the distances from our steady-state FRET experiments using the Förster equation (*Stryer and Haugland, 1967*) and superimposed them on the distributions from chiLife (*Figure 4B*, vertical solid black and red lines). Whereas the distance changes based on our tmFRET measurements were generally similar to the structural predictions, the absolute distances differed by as much as 5 Å. Steady-state tmFRET measurements combine several sources of heterogeneity from the protein into a single weighted average, including the presence of mixed populations of resting and active states. An additional source of uncertainty is the labeling efficiency. If, for example, we assume that 10% of the protein sample in steady-state experiments was not labeled with acceptor (e.g. due to cysteine oxidation), the adjusted weighted-average distances from our experimental data become more similar to predicted peak values (vertical dashed black and red lines). Although overall these average distances are close to the expected distances predicted by the structures, a method for measuring the distance distributions would be desirable.

## Time-resolved tmFRET for determining distance distributions

While steady-state tmFRET shows an *average* FRET efficiency (and weighted-average distance) for all the molecules in the sample, time-resolved measurements of fluorescence lifetimes can be used to measure distance *distributions* in the sample. This approach can account for the two forms of heterogeneity in the protein mentioned earlier (within state and between state heterogeneity), as well as protein not labeled with acceptor. Here, we measured fluorescence lifetimes in the frequency-domain, which can obtain distance distributions utilizing the same principles discussed above for time-domain lifetime measurements (*Figure 1D*). Instead of measuring donor emission photons in response to an impulse of excitation light, lifetimes in the frequency-domain quantify the phase shift (phase delay) and decrease in amplitude of response (modulation ratio) of the donor emission as functions of the modulation frequency of a sinusoidally-varying excitation light (*Zagotta et al., 2024*).

We first measured the fluorescence lifetime of SthK$_{Cterm}$-S361Acd-V416C in the absence of acceptor. The phase delay and modulation ratio as a function of the modulation frequency of the excitation light are shown as gray symbols in the Weber plot of *Figure 5A*. We fit these data with a model for a single exponential lifetime with a time constant of 17.2±0.01 ns (n=8, *Figure 5A*, grey curves), comparable but a little longer than values seen previously for Acd (~16 ns; *Zagotta et al., 2024*; *Speight et al., 2013*). When [Fe(phenM)$_3$]$^{2+}$ was added to the protein sample, the average fluorescence lifetime decreased, illustrated by a shift in the phase delay and modulation ratios to higher frequencies in the Weber plot (*Figure 5A*, black circles). Adding a saturating concentration of cAMP (1.23 mM), further decreased the average lifetime, indicating increased FRET (*Figure 5A*, red circles). This change in lifetime is consistent with steady-state measurements and reflects a decreased distance between the C-helix and the β-roll. In contrast, adding a saturating concentration of cGMP (1.23 mM) instead of cAMP only moderately decreased the fluorescence lifetime, with phase delay and modulation ratio curves falling closer to the apo conditions (*Figure 5A*, green circles).

To further validate our lifetime measurements using a different acceptor with a different R$_0$, we repeated these experiments using [Ru(bpy)$_2$phenM]$^{2+}$ as the acceptor instead of [Fe(phenM)$_3$]$^{2+}$.

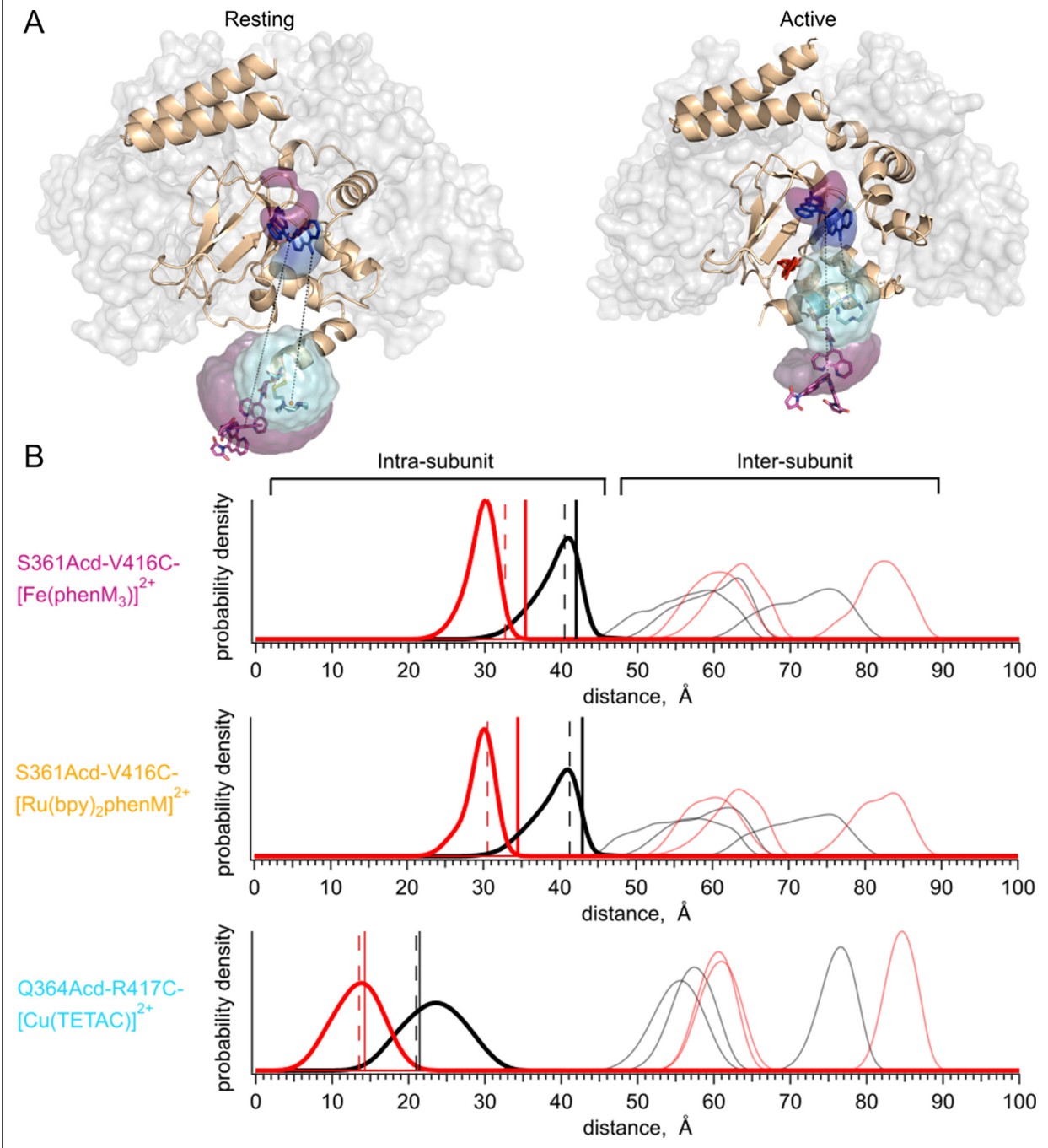

**Figure 4.** Distance distribution predictions with chiLife. (**A**) Structure of resting state (left, PDB:7RSH) (**Gao et al., 2022**), and cAMP-bound active state (right, PDB:4D7T) (**Kesters et al., 2015**) with rotameric clouds predicted by chiLife for the labels (Acd, blue; [Fe(phenM)$_3$]$^{2+}$, magenta; and [Cu(TETAC)]$^{2+}$, cyan). Gray surface indicates location of adjacent WT subunits in tetramer. (**B**) Distance distributions predicted by chiLife for SthK$_{Cterm}$-S361Acd-V416C-[Fe(phenM)$_3$]$^{2+}$, SthK$_{Cterm}$-S361Acd-V416C-[Ru(bpy)$_2$phenM]$^{2+}$ and SthK$_{Cterm}$-Q364Acd-R417C-[Cu(TETAC)]$^{2+}$ (intra-subunit distances: resting, black curves; active, red curves; and inter-subunit distances: resting, grey curves; active, pink curves). Average steady-state tmFRET distance measurements are overlaid as vertical lines for data from **Figure 3** (apo, solid black, and cAMP, solid red) and adjusted assuming a 10% unlabeled protein (dashed black and red vertical lines).

The online version of this article includes the following source data for figure 4:

**Source data 1.** Excel data for chiLife distance distributions (**Figure 4B**).

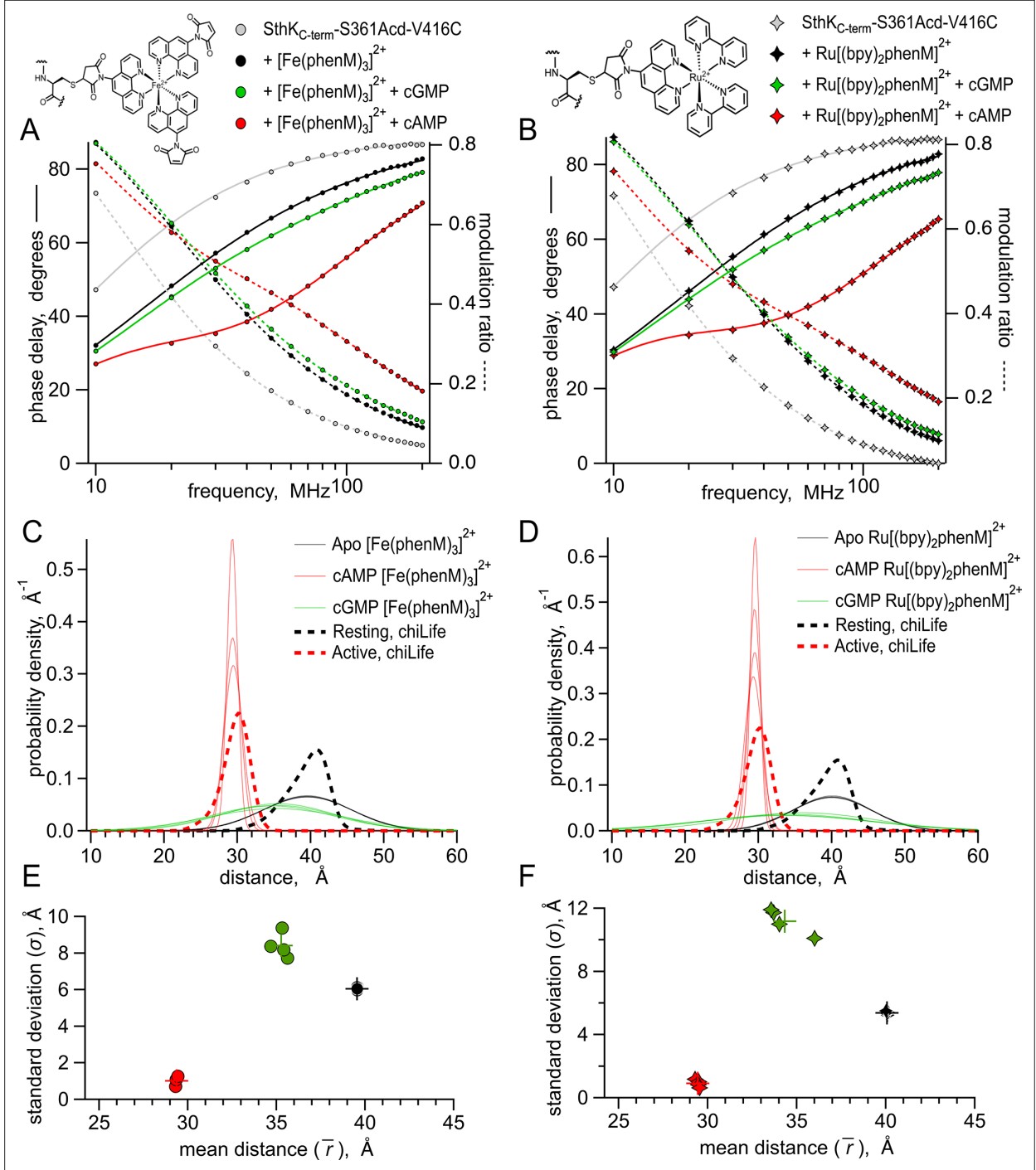

**Figure 5.** Lifetime measurements of SthK$_{Cterm}$-S361Acd-V416C with [Fe(phenM)$_3$]$^{2+}$ and [Ru(bpy)$_2$phenM]$^{2+}$. Chemical structures of acceptors and legends for all plots shown on top. (**A–B**) Representative Weber plots of phase delay and modulation ratio for SthK$_{Cterm}$-S361Acd-V416C labeled with [Fe(phenM)$_3$]$^{2+}$ (**A**) and [Ru(bpy)$_2$phenM]$^{2+}$(**B**). Fits of the data using the single Gaussian model are shown with phase delay as solid curves and modulation ratio as dashed curves. (**C–D**) Spaghetti plots showing distance distributions from the model fits with apo (thin black curves, n=4), with 1.23 mM cAMP (thin red curves, n=4) and 1.23 mM cGMP (thin green curves, n=4) for [Fe(phenM)$_3$]$^{2+}$ (**C**) and [Ru(bpy)$_2$phenM]$^{2+}$(**D**). Distributions predicted by chiLife are overlaid in dashed curves. (**E–F**) Summary of Gaussian fit standard deviations, $\sigma$, versus average distances, $\bar{r}$, for [Fe(phenM)$_3$]$^{2+}$ (**E**) and [Ru(bpy)$_2$phenM]$^{2+}$ (**F**), with average values as cross marks.

The online version of this article includes the following source data and figure supplement(s) for figure 5:

**Source data 1.** Excel data for Weber plots, spaghetti plots and summary of Gaussian fit scatter plots (*Figure 5A–F*).

*Figure 5 continued on next page*

*Figure 5 continued*

**Figure supplement 1.** Representative Weber plot of measured lifetimes of SthK$_{Cterm}$-S361Acd donor-only (grey), then in the presence of [Fe(phenM)$_3$]$^{2+}$ (black), and after the addition of 1.23 mM cAMP (red), showing no change in lifetimes in the absence of a cysteine residue.

**Figure supplement 1—source data 1.** Excel data for fluorescence lifetimes Weber plot.

**Figure supplement 2.** Parameters used in the lifetimes fitting model for both the single and sum of two Gaussian approaches, shown in blue.

[Ru(bpy)$_2$phenM]$^{2+}$ also decreased the average fluorescence lifetime in all three conditions (apo, cAMP and cGMP) relative to the donor-only lifetime (*Figure 5B*). As expected from the longer R$_0$ value for Acd-[Ru(bpy)$_2$phenM]$^{2+}$ compared to Acd-[Fe(phenM)$_3$]$^{2+}$, [Ru(bpy)$_2$phenM]$^{2+}$ produced an even greater shift in phase delays and modulation ratios towards higher frequencies compared to [Fe(phenM)$_3$]$^{2+}$ (*Figures 5B and 6B*). Neither [Fe(phenM)$_3$]$^{2+}$ nor [Ru(bpy)$_2$phenM]$^{2+}$ produced a change in lifetime for SthK$_{Cterm}$-S361Acd (without the cysteine) used as a negative control (*Figure 5—figure supplement 1*). The decreases in lifetimes of SthK$_{Cterm}$-S361Acd-V416C with the metal acceptors and ligands reflect measurable decreases in the donor-acceptor distances (i.e. conformational state) with the addition of cyclic nucleotide, as seen with the steady-state tmFRET measurements.

To obtain distance distributions and average distances that reflect individual conformational states, we fit our lifetime data to a previously described model that assumes each ligand condition (apo, cAMP, and cGMP) is described by a single Gaussian distribution of distances (parameters in *Figure 5— figure supplement 2*; *Zagotta et al., 2021*). The [Fe(phenM)$_3$]$^{2+}$ and [Ru(bpy)$_2$phenM]$^{2+}$ time-resolved tmFRET datasets for each ligand condition were individually fit using this lifetime model (solid and dashed curves, *Figure 5A and B*; *Zagotta et al., 2021*; *Zagotta et al., 2024*). The average distances ($\bar{r}$) and standard deviations ($\sigma$) of the Gaussian distance distributions from these fits are shown for [Fe(phenM)$_3$]$^{2+}$ and for [Ru(bpy)$_2$phenM]$^{2+}$ as spaghetti plots experiments in *Figure 5C and D*, respectively. Overlayed on the spaghetti plots are the distance distributions for the resting and active states predicted using chiLife (dashed curves). The $\bar{r}$ and $\sigma$ of the lifetime model from individual experiments for apo, cAMP and cGMP are summarized in *Figure 5E and F*. Average Gaussian distances measured from experiments for apo and cAMP using both [Fe(phenM)$_3$]$^{2+}$ (apo=39.6 Å, cAMP=29.4 Å) and [Ru(bpy)$_2$phenM]$^{2+}$ (apo=40.1 Å, cAMP=29.4 Å) agreed remarkably well with the chiLife predictions and with each other despite the difference in R$_0$ between [Fe(phenM)$_3$]$^{2+}$ and [Ru(bpy)$_2$phenM]$^{2+}$. Interestingly, the cGMP data had an $\bar{r}$ distance between the apo and cAMP $\bar{r}$ values and had a much wider $\sigma$, spanning a distance range of the apo and cAMP Gaussians combined.

While this lifetime model, with a single average distance for each ligand condition, fits the time-resolved tmFRET data well, it seems likely that there might be more than one conformational state (resting and active) present for each ligand condition (*DeBerg et al., 2016*). As our four-state model suggests, each liganded condition (apo, cAMP, and cGMP) should be comprised of a mixture of resting and active conformational states at varying proportions. For example, the wide cGMP Gaussian positioned between those of the apo and cAMP distributions might reflect the sum of two Gaussians, one with a longer average distance (resting conformation) and one with a shorter average distance (active conformation). The larger apo state $\sigma$ compared to the chiLife prediction could also be due to the presence of a small fraction of the active state, even without ligand. Determining the fractional occupancy among the resting and active states in the apo, cAMP and cGMP conditions would provide energetic information about the transition between resting and active SthK$_{Cterm}$ in the different conditions.

## Global fitting [Fe(PhenM)$_3$]$^{2+}$ and [Ru(Bpy)$_2$PhenM]$^{2+}$ for energetic information

Our approach to measuring fluorescence lifetimes also provides a model-independent way to visually estimate the probability distribution among resting and active states using a representation of the data known as a phasor plot. Phasor plots show the in-phase (*D*) and out-of-phase (*N*) components that underlie the phase delays and modulation ratios shown in the Weber plots (*Digman et al., 2008*). The location of the data on the phasor plot allows us to view complex lifetimes without assumptions about the shape of the distance distributions.

The phasor plot for SthK$_{Cterm}$-S361Acd-V416C modified by [Fe(phenM)$_3$]$^{2+}$ is shown in *Figure 6A*. Single-exponential fluorescence lifetimes fall on the universal circle in phasor plots, and for donor-only

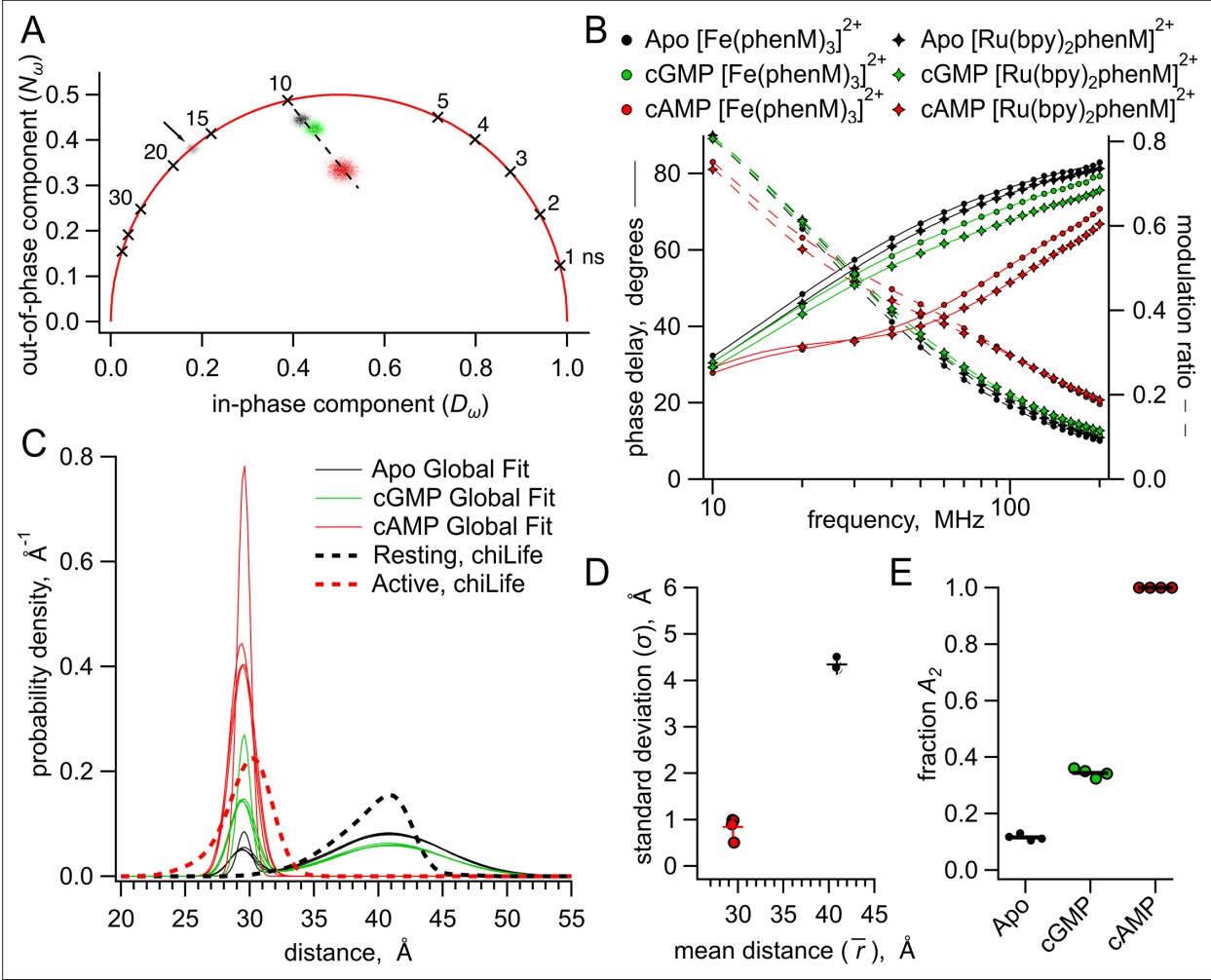

**Figure 6.** Analysis of lifetime data with global fit model allowing sum of two Gaussian distance distributions. (**A**) Representative phasor plot of measured lifetimes with $[Fe(phenM)_3]^{2+}$ acceptor, where markers on universal circle indicate single-exponential time constants (in nanoseconds). Data shown are donor-only (grey), apo (black), cAMP (red) and cGMP (green). (**B**) Representative Weber plot showing global fits for $[Fe(phenM)_3]^{2+}$ acceptor data (closed circles) and $[Ru(Bpy)_2phenM]^{2+}$ acceptor data (open diamonds) for apo (black), cAMP (red), and cGMP (green) conditions. (**C**) Spaghetti plot of distance distributions for each experiment (n=4) (thin lines). For comparison, chiLife distributions are overlayed (dashed curves). (**D**) Summary of Gaussian fit standard deviations, $\sigma$, versus average distances, $\bar{r}$, for apo and cAMP, with average values as cross marks. Colors correspond to conditions in (**A–C**). (**E**) Fit values and averages for fraction activation ($A_2$) for each condition.

The online version of this article includes the following source data and figure supplement(s) for figure 6:

**Source data 1.** Excel data for phasor plot coordinates, Weber plot, spaghetti plot and dot plot for summary of Gaussian fit parameters from global fits (*Figure 6A–E*).

**Figure supplement 1.** Representative phasor plot of measured lifetimes of SthK$_{Cterm}$-S361Acd-V416C alone (donor-only, grey), in the presence of $[Ru(bpy)_2phenM]^{2+}$ (apo, black), in the presence of 1 µM cAMP (cyan) and presence of 1.23 mM cAMP (red).

**Figure supplement 1—source data 1.** Excel data for phasor plot coordinates.

**Figure supplement 2.** Identifiability of parameters in the sum of two Gaussian distributions model, with global fitting $[Fe(phenM)_3]^{2+}$ and $[Ru(bpy)_2phenM]^{2+}$ data.

**Figure supplement 2—source data 1.** Excel data for $\chi^2$ surface graphs for different model parameters (*Figure 6—figure supplement 2A–D*).

SthK$_{Cterm}$-S361Acd-V416C, the lifetime data fell on the universal circle at 17 ns (*Figure 6A*, grey marked with arrow). When fluorescence lifetimes are multi-exponential or nonexponential, the decay data fall inside the universal circle. The data from apo, cAMP, and cGMP conditions all reside within the circle reflecting their complex mixtures of distances. A saturating concentration of cGMP (*Figure 6A*, green) gives data that lay on a line connecting the apo data (*Figure 6A*, black) and the cAMP data (*Figure 6A*, red). This is expected if the cGMP data arises from a mixture of the same distributions found in the

apo and cAMP conditions. In contrast, if cGMP were to produce a distinct conformational state with its own donor-acceptor distance, the data would be predicted to lie off the line connecting apo and saturating cAMP concentrations. Using subsaturating concentrations of cAMP illustrates this point as these data also lay along the line connecting apo and saturating cAMP (*Figure 6—figure supplement 1*). Based on these results, we conclude that the active state produced by cGMP is structurally similar to that produced by cAMP, which is consistent with the previous X-ray crystal structures of C-terminal SthK with cAMP and cGMP (*Kesters et al., 2015*). Although the active state is the same, there must be a higher fraction of protein in the resting state in the presence of cGMP compared to cAMP.

To quantify the fraction of resting and active state in apo, cAMP, and cGMP, we next analyzed our lifetime data by globally fitting them to the lifetime model for Gaussian distributions of distances (*Figure 5—figure supplement 2*). The determination of distance distributions from lifetime experiments alone is an ill-posed problem (*Lakowicz, 2006*). Using global fits of multiple data sets using three different conditions (apo, cAMP, and cGMP) and two different acceptors with different $R_0$s increases our ability to determine the various free parameters. For our global fitting analysis, we assumed that the acceptor complexes $[Fe(phenM)_3]^{2+}$ and $[Ru(bpy)_2phenM]^{2+}$ would produce the same measured distance distribution (both within states and between states) for the same acceptor site. This is a reasonable assumption given that: (1) the chemical structures of the complexes are very similar (*Figure 2B*); (2) chiLife modeling of these two different acceptors produced nearly identical distance distributions (*Figure 4B*); and (3) dose-response curves for cAMP measured using steady-state FRET were comparable for $[Fe(phenM)_3]^{2+}$- and $[Ru(bpy)_2phenM]^{2+}$-modified protein (*Figure 3D*). Parameter identifiability for this lifetime model was observed in graphs of $\chi^2$ minimized curves as functions of fixed ranges of each parameter, where minima for each parameter were resolved (*Figure 6—figure supplement 2*). This global fitting allowed us to determine the fraction of resting and active components, and therefore the free energy change, in each condition.

We globally fit six data sets (each set included phase delay and modulation ratio data), representing the conditions of apo, cAMP, and cGMP with the two different acceptors $[Fe(phenM)_3]^{2+}$ and $[Ru(bpy)_2phenM]^{2+}$. We parameterized the distance distribution as the sum of two Gaussians. Each of the Gaussian parameters $\bar{r}_1$, $\bar{r}_2$, $\sigma_1$, and $\sigma_2$ were assumed to be the same across all conditions, and the fraction of each Gaussian (set by parameter $A_2$) was allowed to vary between conditions. Global fitting of the lifetime model provided an excellent fit to all of the data (*Figure 6B*) revealing the distance distributions shown in *Figure 6C*. Similar to the chiLife predictions (*Figure 4*) and the single Gaussian fits (*Figure 5*), the global fits gave $\bar{r}_1$ and $\bar{r}_2$ values of 40.9 Å and 29.5 Å in the resting and active states, respectively, with $\sigma_1$ and $\sigma_2$ values of 4.3 Å and 0.84 Å in the resting and active states, respectively (*Figure 6C and D*). Surprisingly, our fits indicated that the apo condition is best explained with 12% in the active state ($A_2$=0.12) and 88% in the resting state, whereas the saturating cAMP concentration condition was best fit with only the active state ($A_2$=1) (*Figure 6E*). In contrast, cGMP was fit with an $A_2$ of 0.34. Given that the qualities of these fits are comparable to the single Gaussian fits (with similar $\chi^2$ values), and fewer parameters were used in the global fits than across the sum of comparable data sets with single Gaussian fits, we find the global fitting approach more parsimonious with the phasor plot data and previous SthK X-ray crystallography data in cGMP (*Kesters et al., 2015*).

**Table 1.** Calculated energetics of the four state model.

| Ligand condition | Ionic strength (mM KCl) | ΔG (kcal/mol) | ΔΔG (kcal/mol) |
|---|---|---|---|
| | 150 | 1.14±0.03 | -- |
| Apo | 500 | 1.2±0.03 | -- |
| | 150 | -- | -- |
| cAMP | 500 | <–2.7±0 | <–3.9±0.03 |
| | 150 | 0.67±0.03 | –0.47±0.04 |
| cGMP | 500 | 0.38±0.02 | –0.82±0.04 |

Energies are reported in mean values ± SEM.

Using the probabilities of resting and active states in each condition and the Gibbs free energy equation, we calculated the $\Delta G$ and $\Delta\Delta G$ energetics of the transitions in the four-state diagram (*Figure 1C* and *Table 1*). In the absence of cyclic nucleotide, the lifetime model fit gave an $A_2$ of 0.12, corresponding to a $\Delta G_{Apo}$ of 1.2 kcal/mol. The isolated SthK$_{Cterm}$, therefore, exhibits considerable activation even in the apo state. For saturating cAMP, for which the global fit indicated an $A_2$ of 1, we assumed a maximum contribution of the resting state of 0.01, yielding a $\Delta G_{cAMP}$ that is more favorable than –2.7 kcal/mol and a $\Delta\Delta G_{cAMP}$ that is more favorable than –3.9 kcal/mol. With a $\Delta\Delta G_{cGMP}$ of –0.82 kcal/mol, cGMP was found here to be a partial agonist, consistent with electrophysiology and previous experiments (*Morgan et al., 2019*; *Schmidpeter et al., 2018*; *Rheinberger et al., 2018*). Interestingly, our measurements reporting the conformational change in the isolated CNBD show a greater activation for apo, cAMP and cGMP, than electrophysiology experiments on full-length channels. These results demonstrate that time-resolved tmFRET can be utilized to obtain energetic information on the individual domains during the allosteric activation of SthK.

## Changes in energetics with salt concentrations

We were surprised by the high probability of being in the active state for cGMP ($A_2$=0.34) indicated by the sum of Gaussian fits (*Figure 6E*), especially when electrophysiology shows a much lower open probability with cGMP (*Figure 1A*). Although this difference might be due to intrinsic energetic differences between our fragment SthK$_{Cterm}$ construct and the full-length channel, it could also arise, in part, from the high ionic conditions (500 mM) used in our lifetime experiments compared with physiological ionic concentrations used in electrophysiology experiments. Known inter-subunit C-terminal salt bridge interactions could be destabilized by increased ionic strength and, previously, it was observed that higher ionic strength eliminated tetramerization in HCN1 C-terminal fragments (*Morgan et al., 2019*; *Lolicato et al., 2011*; *Craven and Zagotta, 2004*). However, we found no difference in elution volume for SEC experiments performed with SthK$_{Cterm}$-S361Acd-V416C using 150 mM instead of 500 mM KCl, indicating that the oligomerization state was the same under both experimental conditions (*Figure 7A*).

To test whether the high ionic strength of our experiments contributed to the high $A_2$ for cGMP, we repeated the above lifetime experiments with a more physiological salt concentration (150 mM KCl). We made new time-resolved tmFRET measurements with SthK$_{Cterm}$-S361Acd-V416C in lower ionic conditions, for both the [Fe(phenM)$_3$]$^{2+}$ and [Ru(bpy)$_2$phenM]$^{2+}$ acceptors across apo, cAMP, and cGMP conditions. Comparison of [Fe(phenM)$_3$]$^{2+}$ lifetime data in the two different ionic conditions are shown in the representative Weber plot in *Figure 7B*. While the apo and cAMP data appeared to be equivalent between 150 and 500 mM KCl, the cGMP data were different between the conditions. When an additional 300 mM KCl was added to protein samples in 150 mM KCl after cGMP application, the phase delay and modulation ratio approached the same lifetimes as the 500 mM KCl cGMP condition (*Figure 7B*, dark green circle), recapitulating the 500 mM KCl data set. We then globally fit the 150 mM KCl data across [Fe(phenM)$_3$]$^{2+}$ and [Ru(bpy)$_2$phenM]$^{2+}$ acceptors for apo, cAMP, and cGMP conditions with the sum of two Gaussians (*Figure 7C*) and found that the average distances, $\bar{r}$, and standard deviations, $\sigma$, were nearly identical to those at 500 mM KCl (*Figure 7D*). The $A_2$ from cGMP fits decreased in 150 mM KCl compared to 500 mM KCl indicating that a decrease in ionic concentration made activation by cGMP less favorable ($\Delta\Delta G_{cGMP}$=–0.47 kcal/mol, $\Delta\Delta\Delta G_{cGMP\ KCl}$=0.35 kcal/mol). In the absence of ligand, $A_2$ was unchanged between the two ionic concentrations (*Figure 7D*), indicating that the increased ionic strength effected only the active conformation in the presence of cGMP. Since activation is already very favorable in the presence of cAMP, we could not determine if ionic strength also effected $\Delta\Delta G_{cAMP}$. These results suggest that ionic strength might specifically affect an electrostatic interaction between CNBD and the cyclic nucleotide during the activation transition.

## Discussion

In this study, we measured both steady-state and time-resolved tmFRET for a C-terminal fragment of SthK and interpreted the data using a four-state allosteric model. This approach allowed us to acquire structural information and energetics of the conformational change of the C-helix under different ligand and experimental conditions. Our findings revealed a small presence of the active state in the absence of ligand, while saturating concentrations of cAMP exhibited energetics consistent with that

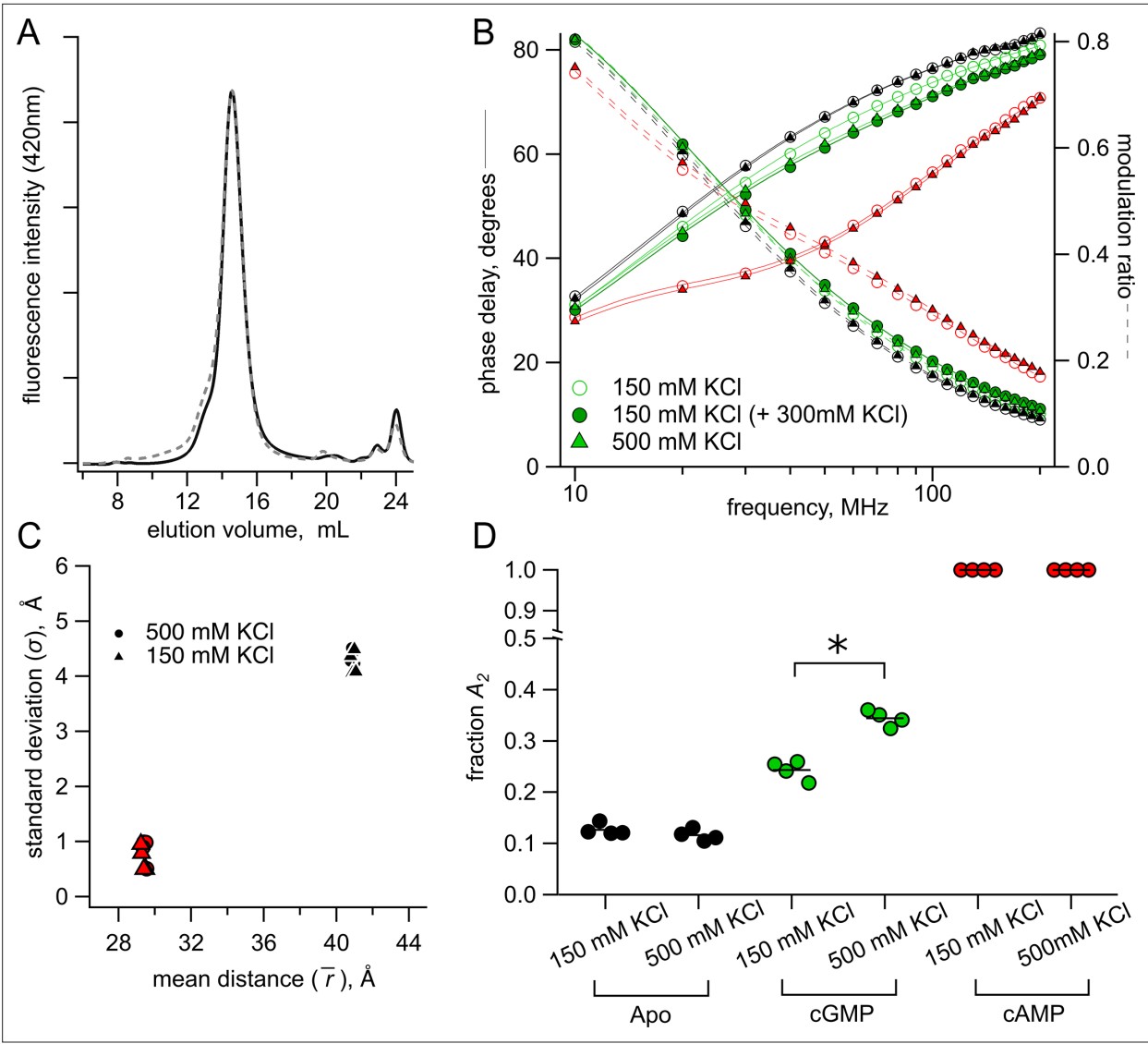

**Figure 7.** Lifetime measurements in 150 mM KCl versus 500 mM KCl conditions. (**A**) Normalized SEC traces of SthK$_{Cterm}$ in 150 mM KCl (solid curve) and 500 mM KCl (dashed curve). (**B**) Representative Weber plot for [Fe(phenM)$_3$]$^{2+}$ lifetimes in 150 mM (open circles) and 500 mM KCl (closed triangles), in each ligand condition (apo, black; cGMP, green; cAMP; red). An additional lifetime trace is shown for an experiment where additional 300 mM KCl was added to the protein sample with 150 mM KCl in cGMP (closed green circles). (**C**) Comparison of Gaussian average distance, $\bar{r}$, and standard deviation, $\sigma$, between ionic concentrations using model global fitting of [Fe(phenM)$_3$]$^{2+}$ and [Ru(bpy)$_2$phenM]$^{2+}$ with the sum of two Gaussians (apo, black, and cAMP, red, n=4). (**D**) Fraction of active state in each condition (A$_2$) for 150 mM KCl and 500 mM KCl using the sum of two Gaussian model fits (apo p=0.2, cGMP *p=0.0001).

The online version of this article includes the following source data for figure 7:

**Source data 1.** Excel data for size exclusion chromatography traces, Weber plot, and dot plots for summary of Gaussian fit parameters from global fits (*Figure 7A–D*).

of a full agonist by strongly shifting occupancy into the active state. In contrast, saturating concentrations of cGMP demonstrated characteristics of a partial agonist, eliciting a lower fraction of the same active conformation observed with cAMP. Furthermore, our results demonstrated the impact of both oligomerization and ionic strength on the energetics of the conformational change of isolated SthK$_{Cterm}$.

Our ability to recover structural information in the form of average distance, $\bar{r}$, and standard deviation of distances, $\sigma$, from time-resolved tmFRET indicates that this approach is superior to weighted-average steady-state distance measurements alone. The average distance values obtained from

lifetime measurements closely matched the distances predicted by chiLife, indicating that the characterized states likely correspond to the observed resting and active states in existing X-ray and cryo-EM structures (*Kesters et al., 2015*; *Gao et al., 2022*). Whereas the active state's standard deviation ($\sigma_2$) closely aligned and was slightly narrower than the chiLife predictions, the resting state's standard deviation ($\sigma_1$) was larger than chiLife predictions, even with the global fitting approach. This indicates a broad structural heterogeneity in the resting state that cannot be explained by the rotamer clouds of the labels alone, such as heterogeneity in the position of the C-helix backbone, which is consistent with several past experimental observations (*Rheinberger et al., 2018*; *Taraska et al., 2009*; *Matulef and Zagotta, 2002*). Overall, time-resolved tmFRET reliably provides valuable, although sparse, structural insights, especially in cases of heterogeneity where current structural methods are limited.

In addition to the heterogeneity within a given conformational state, time-resolved tmFRET allowed us to obtain the heterogeneity between conformational states ($A_2$) and thus energetics of state transitions in our four-state model. We determined the distribution between these four conformational states by employing a model comprising the sum of two Gaussian distributions, and globally fitting data across distinct acceptors and ligand conditions. This analysis allowed us to quantify the changes in free energy ($\Delta G$) describing our four-state model. Additionally, we were able to calculate the $\Delta\Delta G$ for each ligand, which represents the free energy imparted by the ligand to drive the conformational change of the CNBD. Although we assumed a model with only two stable conformational states, this may not capture all the stable states in the CNBD. Even if the conformational distributions assumed here were oversimplified, our time-resolved tmFRET approach still allowed for a deeper understanding of the allostery in the C-terminal of SthK.

There are some complications with our time-resolved tmFRET approach that should be considered in future applications. (1) Donor and acceptor sites for tmFRET were carefully selected for solvent accessibility, but it is possible that the introduction of Acd and metal acceptor altered the energetics compared to WT SthK$_{Cterm}$. (2) Incomplete labeling in our protein samples introduces a fraction of donor-only molecules, which are not captured in steady-state measurements but are estimated in our lifetime fits. We have previously shown that there is some degree of correlation among parameters in our FRET model, particularly between the fraction of donor-only and the standard deviation of the Gaussians (*Zagotta et al., 2024*). As a result, it is important to ensure a low contribution (<15%) from the donor-only fraction, and this fraction should be validated by independent methods. (3) Although parameters all reliably converged to $\chi^2$ minimized values in our global fits, some parameters were more identifiable than others (*Figure 6—figure supplement 2*). (4) Both the single Gaussian distribution model and the sum of two Gaussian distributions model yielded minimized $\chi^2$s and excellent fits; however, our results do not conclusively favor one lifetime model over the other. Whereas the single Gaussian distribution fits contain fewer assumptions, the sum of two Gaussian global fits contain fewer free parameters across the multiple acceptors and conditions. We believe that the assumptions in the four-state model and sum of two Gaussians fitting are reasonable based on both previous experiments and theoretical considerations (*Morgan et al., 2019*; *DeBerg et al., 2016*; *Zagotta et al., 2024*).

A number of previous studies have measured the energetics of allostery in CNBD channels; however, interpreting our results within the context of these previous studies presents challenges. These studies have utilized various different CNBD channels where the energetics, including even the ligand specificity, are very different (*Morgan et al., 2019*; *DeBerg et al., 2016*; *Collauto et al., 2017*; *Craven and Zagotta, 2004*; *Puljung et al., 2014*; *Evans et al., 2020*; *Goldschen-Ohm et al., 2016*; *Pfleger et al., 2021*; *Craven et al., 2008*; *Kondapuram et al., 2022*). For example, in CNGA1 channels from rod photoreceptors, cGMP is a full agonist, and cAMP is a weak partial agonist, while in SthK the agonist specificity is reversed (*Brams et al., 2014*; *Morgan et al., 2019*; *Gordon and Zagotta, 1995*; *Varnum et al., 1995*). In addition, most of these previous studies have been limited to electrophysiology, which measures the open probability of the pore. Without additional information, the open probability provides only an indirect measurement of the energetics of other domains like the ligand-binding domain. While double electron-electron resonance spectroscopy (DEER; *DeBerg et al., 2016*; *Collauto et al., 2017*; *Puljung et al., 2014*; *Evans et al., 2020*) and single-molecule fluorescence studies (*Gao et al., 2022*; *Goldschen-Ohm et al., 2016*; *Goldschen-Ohm et al., 2017*) offer the potential to measure the energetics of individual domains, these methods have not yet been applied to the SthK CNBD. Interestingly, DEER spectroscopy on the HCN2 CNBD fragment revealed a mixture of resting and active states in cAMP (*DeBerg et al., 2016*), whereas we observed

in the SthK CNBD fragment a more favorable transition in cAMP. These differences might be due to either inherent differences in the energetics between the C-terminal domains of SthK and HCN2, or the different experimental conditions used, such as the monomeric protein used in the DEER studies vs the tetrameric C-terminal domain studied here. Although their energetics may differ, all CNBD channels studied to date appear to undergo an allosteric transition of the CNBD where the binding of cyclic nucleotide promotes a movement of the C-helix relative to the β-roll that is coupled indirectly to the opening of the pore.

For simplicity, here we used the C-terminal fragment of SthK instead of the full-length channel. The relationship between the energetics of the C-terminal fragment and the full-length channel is likely complex and requires further study. For example, coupling of the conformational changes in the individual CNBDs to a concerted opening of the pore is expected to alter both the ΔG for the CNBD transition and the cooperativity of the transitions between subunits. This coupling would also make the ΔΔG measured in the pore by electrophysiology smaller than the ΔΔG we measured in the CNBD. If the conformational change in the CNBD is the same in the fragment and full-length channel, as suggested by structural studies (*Kesters et al., 2015*; *Gao et al., 2022*), we could use our measured ΔΔG from the fragment, together with measurements from the full-length channel, to develop a full allosteric model of the intact channel.

# Materials and methods

## Key resources table

| Reagent type (species) or resource | Designation | Source or reference | Identifiers | Additional information |
|---|---|---|---|---|
| Strain, strain background (*Escherichia coli*) | B-95.ΔA *E. coli* | Addgene | Bacterial strain #197933 | |
| Recombinant DNA reagent | pDule2-Mj Acd A9 (plasmid) | Addgene | Plasmid #197652 | |
| Recombinant DNA reagent | MBP-S361TAG -ctermSthK-TEV-TWS.pETM11 (plasmid) | Addgene | Plasmid #231098 | |
| Recombinant DNA reagent | MBP- S361TAG -V416C-ctermSthK-TEV-TWS.pETM11 (plasmid) | Addgene | Plasmid #231099 | |
| Recombinant DNA reagent | MBP-Q364TAG -ctermSthK-TEV-TWS.pETM11 (plasmid) | Addgene | Plasmid #231100 | |
| Recombinant DNA reagent | MBP- Q364TAG -R417C-ctermSthK-TEV-TWS.pETM11 (plasmid) | Addgene | Plasmid #231101 | |
| Software, algorithm | FDlifetime_17_Igor_procedures.ipf | *Zagotta, 2021* | https://github.com/zagotta/FDlifetime_program | |

## Constructs and mutagenesis

The C-terminal construct of SthK (SthK$_{Cterm}$) was created by replacing the first 224 residues of cysteine-free SthK in the pETM11 vector (described previously, Uniprot accession #E0R11; *Morgan et al., 2019*), with the sequence for MBP followed by a 14-residue asparagine linker and a TEV protease cleavage site, using restriction digestion and T4 ligation. Following the C-terminal end of truncated SthK sequence (225-430), another TEV cleavage site and a Twin-Strep-tag sequence was incorporated using Gibson cloning. An amber stop codon (TAG) was introduced at positions 359, 361 or 364 of the WT SthK$_{Cterm}$ using site directed mutagenesis (*Liu and Naismith, 2008*) to create new 'donor-only' constructs. To generate constructs with both donor and acceptor sites, a single cysteine mutation was introduced into each donor-only construct at sites 416 or 417, to make the three constructs: SthK$_{Cterm}$-I359TAG-V417C, SthK$_{Cterm}$-S361TAG-V416C, and SthK$_{Cterm}$-Q364TAG-V417C.

## Expression and purification of the C-terminal SthK fragment

The WT SthK$_{C-term}$, donor-only, and cysteine-containing constructs were each co-transformed with the AcdA9 aminoacyl tRNA synthetase/tRNA-containing plasmid (pDule2-Mj Acd A9) (*Sungwienwong et al., 2017*) into B-95.ΔA *E. coli* (DE3) cells (*Mukai et al., 2015*). The transformed *E. coli* cultures were grown in terrific broth medium at 37 °C in 50 μg/ml kanamycin and 60 μg/ml spectinomycin to an OD$_{600}$ ~1.0 before adding Acd (at a final concentration of 0.3 mM), and 0.5 mM isopropyl

β-D-1-thiogalactopyranoside (IPTG) for protein induction (*Speight et al., 2013*; *Wu and Piszczek, 2021*; *Jones et al., 2020*). The cultures were transferred to 18 °C to grow an additional 17–19 hr, cells were harvested by centrifugation, and cell pellets were resuspended in lysis buffer (150 mM KCl, 50 mM Tris, 2 mM β-mercaptoethanol, and 10% glycerol, pH 7.4 supplemented with protease inhibitor tablets, EDTA-free [Pierce, Thermo Fisher]). Cell suspensions were lysed on an Avestin Emul-siFlex-C3 cell disruptor three times at 15,000–20,000 psi, then diluted with lysis buffer and cleared by centrifugation at 39,000 x $g$ for 30 min. Clarified lysate was loaded onto 0.75 mL of Strep-Tactin Superflow high-capacity beads (IBA Biosciences) at 4 °C in a disposable column. The resin was washed with 25 column volumes of KBT solution (150 mM KCl, 50 mM Tris, 10% glycerol, pH 7.9) and eluted with 10 mM d-Desthiobiotin (Sigma) in KBT buffer. Eluted protein was treated with 10 mM tris(2-carboxyethyl)phosphine (TCEP) to fully reduce cysteines, flash frozen with liquid nitrogen, and stored at –80 °C.

Acd-labeled protein was mixed with WT SthK$_{Cterm}$ protein in ratios of >3:1 and were then TEV cleaved with 1:10 vol/vol TEV protease (1.2 mg/ml) for 18 hours at room temperature to cleave the N-terminal MBP and C-terminal Twin-Strep-tag. TEV-cleaved protein was then diluted 1:500 in 10 mM KCl, 50 mM Tris, 10% glycerol, pH 7.9 and then loaded onto a HiTrap Q HP column (GE Healthcare) for ion exchange chromatography (IEC) to remove TEV protease, and cleavage fragments from SthK$_{Cterm}$. SthK$_{Cterm}$ was eluted with an increasing concentration ramp of KCl (10 mM – 1 M) and fractions were collected, with SthK$_{Cterm}$ coming off in a single peak. We used size exclusion chromatography (SEC) on a Superdex 200 10/300 column (GE Healthcare) to evaluate oligomerization state or collect fractions in either KBT or High-KBT (500 mM KCl, 50 mM Tris, 10% Glycerol, pH 7.9).

## Mass photometry

Mass photometry utilizes light scattering to measure the molecular weight of proteins along with their relative abundance in a sample, with an analytical range of 40 kDa - 5 MDa (*Young et al., 2018*; *Lako-wicz et al., 1991*). Glass coverslips (24x50 mm) were isopropanol-cleaned and dried with nitrogen gas. We used 10 nM ß-amylase (BAM) to calibrate a TwoMP Mass Photometer instrument (Refeyen Ltd, Oxford, UK). As a background reference, 10 µl of buffer solution (KBT with 150 mM KCl) was first measured in sample gasket wells. Collected SEC SthK$_{Cterm}$ fractions were then measured (10 µl added for total of 20 µl, ~50 nM) and mass histograms collected via the DiscoverMP analysis software. The mass histograms were analyzed with single Gaussian fits in IGOR-Pro v.8 (Wavemetrics, Lake Oswego, OR).

## Cyclic nucleotides and labeling reagents

Adenosine 3',5'-cyclic monophosphate sodium salt monohydrate (cAMP) and guanosine 3',5'- cyclic monophosphate sodium salt (cGMP) were purchased from Sigma-Aldrich and both prepared at 16 mM in 150 mM KCl, 50 mM Tris and 10% glycerol, pH 7.4. Metal acceptor complexes, [Fe(phenM)$_3$]$^{2+}$, [Ru(bpy)$_2$phenM]$^{2+}$, and [Cu(TETAC)]$^{2+}$ for tmFRET were prepared as previously described (*Gordon et al., 2024*; *Zagotta et al., 2024*).

## SthK$_{Cterm}$ steady-state fluorescence measurements

For steady-state tetramer experiments (*Figure 3*), fractions from IEC-purified SthK$_{Cterm}$ donor-only or SthK$_{Cterm}$-Acd-cysteine protein, with excess WT SthK$_{Cterm}$, were used at high concentration immediately after elution from the IEC. Protein was confirmed as tetrameric by analytical SEC and diluted 1:2 to 1:5 into KBT into quartz cuvettes. Acd fluorescence intensity was measured at 10 s intervals using Jobin Yvon Horiba FluoroMax-3 spectrofluorometer in anti-photobleaching mode (Edison, NJ) as described previously (*Zagotta et al., 2021*; *Gordon et al., 2024*). Acceptor molecules at final concentrations of 6 µM [Fe(phenM)$_3$]$^{2+}$, 20 µM [Ru(bpy)$_2$phenM]$^{2+}$ or 10 µM [Cu(TETAC)]$^{2+}$ were added to the cuvettes followed by 160 µM final concentration of either cAMP or cGMP. For [Cu(TETAC)]$^{2+}$ experiments, 5 mM TCEP was added after cAMP or cGMP to reverse the tmFRET by removing the acceptor. All experiments had a 'no-cysteine' protein control to correct the FRET efficiency for intensity changes without bound acceptor, where FRET efficiency E=1 – (F$_{Cys}$/F$_{No\ Cys}$). In different experiments, FRET efficiencies at increasing concentrations of cAMP were normalized to the maximal observed FRET efficiency and experimental averages were fit with the Hill equation. Fit slopes, $h$, were fixed at 1 for the [Ru(bpy)$_2$phenM]$^{2+}$ and [Cu(TETAC)]$^{2+}$ data but was allowed to vary for the [Fe(phenM)$_3$]$^{2+}$, which

did not fit with a slope of 1. For steady-state tmFRET experiments to compare between oligomeric states, monomeric and tetrameric protein were separated and collected by SEC using a Superdex 75 Increase 10/300 GL column (GE Healthcare) and then used directly in fluorescence experiments (*Figure 3—figure supplement 1*).

## Measurements of fluorescence lifetimes using FLIM

For lifetime studies, tetrameric SthK$_{Cterm}$-S361Acd-V416C protein, mixed >1:3 with WT SthK$_{C-term}$, was labeled with either [Fe(phenM)$_3$]$^{2+}$ or [Ru(bpy)$_2$phenM]$^{2+}$. For [Fe(phenM)$_3$]$^{2+}$ experiments, SthK$_{Cterm}$-S361Acd-V416C protein was buffer exchanged using a BioSpin6 Mini column (BioRad) into KBT (150 mM KCl) or High-KBT (500 mM KCl) then labeled immediately prior to lifetime measurements. For [Ru(bpy)$_2$phenM]$^{2+}$ experiments, SthK$_{Cterm}$-S361Acd-V416C protein was incubated with 1 mM [Ru(bpy)$_2$phenM]$^{2+}$ for 30 min at room temperature and then buffer exchanged into KBT or High-KBT to remove the unincorporated label.

The theory of time-resolved FRET and the implementation of the lifetime FRET model in the frequency domain have been described previously (*Lakowicz, 2006*; *Cheung et al., 1991*; *Lakowicz et al., 1988*; *Hochstrasser et al., 1992*; *Kawate and Gouaux, 2006*). Frequency domain fluorescence lifetime data were obtained using a Q2 laser scanner and A320 FastFLIM system (ISS, Inc, Champaign, IL, USA) attached to a Nikon TE2000U microscope. Acd was excited using a 375 nm pulsed diode laser (ISS, Inc) driven by FastFLIM at 10 MHz, with a long-pass dichroic mirror (387 nM) and a band-pass emission filter (451/106 nm). Collected emission was recorded on a Hamamatsu model H7422P PMT detector. Lifetime measurements were calibrated using Atto 425 in water (with a lifetime of 3.6 ns), using the same optical configuration. Protein samples were used at full strength by pipetting 11 µl onto an ethanol-cleaned #1.5 glass coverslip for recording with a Nikon CFI Super Fluor 10x0.5 NA objective. Confocal images (256x256 pixels) were collected with a pinhole of 200 µm and a pixel dwell time of 1ms. Donor-only lifetimes were obtained from SthK$_{Cterm}$-S361Acd-V416C in KBT and High-KBT solutions prior to the addition of [Fe(phenM)$_3$]$^{2+}$ at a final concentration of 76.8 µM. [Ru(bpy)$_2$phenM]$^{2+}$-labeled protein was measured following buffer exchange. Either 1 µl of 16 mM cAMP or 16 mM cGMP was added to acceptor-labeled protein at a final concentration of 1.23 mM. The phase delays and modulation ratios of the fluorescence signal were obtained using VistaVision software from the sine and cosine Fourier transform of the phase histogram H(p), subject to the instrument response function (*Zagotta et al., 2021*).

## Lifetime distance distribution model

The model for obtaining a single Gaussian distribution, or a sum of two Gaussian distributions of distances from lifetimes has been described previously (*Zagotta et al., 2021*; *Zagotta et al., 2024*). Globally fitting lifetime data across [Fe(phenM)$_3$]$^{2+}$ and [Ru(bpy)$_2$phenM]$^{2+}$ experiments and the conditions of apo, cAMP and cGMP, used the same model as previously described (*Figure 5—figure supplement 2* for parameters), but parameters were constrained differently. Parameters defining the Gaussian states included: two unconstrained average distances ($\bar{r}_1$ and $\bar{r}_2$), two standard deviations ($\sigma_1$ and $\sigma_2$) constrained between values of 0.5–10 Å, and one fraction for the proportions between the two Gaussians ($A_2$) varied independently for each of conditions, apo, cAMP, and cGMP, constrained between 0 and 1. Other parameters that were varied independently included: the fraction of donor-only in each respective sample, the fraction of background in each experiment (<5%), and the $t_0$ offset for each experiment. The measured donor-only lifetimes (in KBT and High-KBT) were single exponential, and the time constant was previously measured and held fixed for the global fitting. The acceptor complex $R_0$ values were determined previously (*Gordon et al., 2024*) and held constant at 41.8 Å and 43.5 Å for [Fe(phenM)$_3$]$^{2+}$ and [Ru(bpy)$_2$phenM]$^{2+}$, respectively (*Hofmann, 2023*). For the single Gaussian lifetime fits, the fraction donor only was held constant within experiments from the same labeled protein samples.

## Energetic and Hill equation calculations

ΔG, ΔΔG, and ΔΔΔG are calculated as follows:

$$\Delta G = -RT * ln\left(\frac{P_{active}}{P_{resting}}\right) \qquad (1)$$

$$\Delta\Delta G_{cNMP} = \Delta G_{cNMP} - \Delta G_{apo} \tag{2}$$

$$\Delta\Delta\Delta G_{cGMP\,KCl} = \Delta\Delta G_{cGMP\,150\,mM\,KCl} - \Delta\Delta G_{cGMP\,500\,mM\,KCl} \tag{3}$$

where $R$ is the molar gas constant, $T$ is the absolute temperature (K), and $P_{active}$ and $P_{resting}$ are the proportion of molecules in the active state and resting respectively, and cNMP is cyclic nucleotide (either cAMP or cGMP).

Hill fits (*Figures 1B and 3D*) were obtained using:

$$Normalized\ Response = \frac{1}{1 + \left(\dfrac{K_{1/2}}{[cAMP]}\right)^h} \tag{4}$$

where $K_{1/2}$ is the concentration of cAMP producing half-maximal FRET change and $h$ is the Hill coefficient.

## chiLife predictions

Computational predictions of the possible rotameric positions for the donor and acceptor labels were made with chiLife using the accessible-volume sampling method as previously described (*Zagotta et al., 2024*; *Tessmer and Stoll, 2023*). Acd, [Cu(TETAC)]$^{2+}$, [Fe(phenM)$_3$]$^{2+}$ and [Ru(bpy)$_2$phenM]$^{2+}$ were added as custom labels in chiLife and modeled onto the cryo-EM structure of the full-length SthK closed state (*Gao et al., 2022*; PDB: 7RSH, residues 226–418) and the X-ray crystallography structure of the cAMP bound SthK C-terminal fragment (*Kesters et al., 2015*; PDB: 4D7T). For each donor-acceptor pair, labels were superimposed at indicated residue positions and 10,000 possible rotamers were modeled. Rotamers resulting in internal clashes (<2 Å) were removed and external clashes evaluated as previously described (*Tessmer and Stoll, 2023*). Donor-acceptor distance distributions were calculated between the remaining (~500–2000) label rotamers for each donor-acceptor pair. Inter-subunit distances were calculated as distances between one Acd molecule and the modeled acceptor rotamers on each of the other three subunits.

## Electrophysiology

The cysteine-free construct of full-length SthK (cfSthK) was cloned into a pcGFP vector (*Kawate and Gouaux, 2006*; with YFP instead of GFP) and expressed in *E. coli* C43(DE3) bacteria (*Miroux and Walker, 1996*). A valine to alanine mutation at position 208 was added for increased open channel probability (A208V-cfSthK-YFP; *Morgan et al., 2019*). Spheroplasts were formed from the C43 samples as previously described (*Morgan et al., 2019*). SthK currents were recorded from inside-out spheroplast patches using an Axopatch 200 A amplifier with Patchmaster software (HEKA Elektronik) as previously described (*Morgan et al., 2019*). The pipette and bath recording solutions were both:150 mM KCl, 20 mM MgCl$_2$, 500 mM sucrose, 10 mM HEPES, pH7.4 (*Morgan et al., 2019*). Currents were measured using jumps in the holding potential from 0 mV to +80 mV. Fractional activation for 1 mM cGMP were obtained by comparing the fractional response to 1 mM cAMP in the same patch.

## Statistics and reproducibility

Data values were expressed as mean ± SEM of n independent experiments, unless stated otherwise, and all error bars are ± SEM. Statistical significance (*p<0.05) was determined by using a two tailed Student's t-test.

## Acknowledgements

We thank the Oregon State University GCE4ALL (Center for Genetic Code Expansion for All) for their long-standing collaboration, and Drs. Chloe Jones, James Petersson, and Kyle D Shaffer. (University of Pennsylvania) for excellent technical support. We also thank Dr. Lisa Tuttle (University of Washington) for assistance with Mass Photometry and all members of the SEG and WNZ laboratories for helpful conversations and support. Research reported in this publication was supported by the National Institutes of Health under award numbers R35GM145225 (to SEG), R35GM148137 and R01EY010329 (to WNZ), T32GM008268 and T32EY7031-43 (to PE).

# Additional information

## Funding

| Funder | Grant reference number | Author |
|---|---|---|
| National Institute of General Medical Sciences | R35GM148137 | William N Zagotta |
| National Eye Institute | R01EY010329 | William N Zagotta |
| National Institute of General Medical Sciences | R35GM145225 | Sharona E Gordon |
| National Institute of General Medical Sciences | T32GM008268 | Pierce Eggan |
| National Eye Institute | T32EY7031 | Pierce Eggan |

The funders had no role in study design, data collection and interpretation, or the decision to submit the work for publication.

## Author contributions

Pierce Eggan, Conceptualization, Data curation, Formal analysis, Investigation, Visualization, Methodology, Writing – original draft, Writing – review and editing; Sharona E Gordon, William N Zagotta, Conceptualization, Data curation, Formal analysis, Funding acquisition, Investigation, Visualization, Methodology, Writing – original draft, Writing – review and editing

## Author ORCIDs

Pierce Eggan ⬤ https://orcid.org/0000-0001-6134-4569
Sharona E Gordon ⬤ https://orcid.org/0000-0002-0914-3361
William N Zagotta ⬤ https://orcid.org/0000-0002-7631-8168

Reviewer #1 (Public review): https://doi.org/10.7554/eLife.99854.3.sa1
Reviewer #2 (Public review): https://doi.org/10.7554/eLife.99854.3.sa2
Reviewer #3 (Public review): https://doi.org/10.7554/eLife.99854.3.sa3
Author response https://doi.org/10.7554/eLife.99854.3.sa4

---

# Additional files

## Supplementary files

• MDAR checklist

## Data availability

Source data files have been included with the final manuscript. Code used for data analysis has been posted to GitHub, (copy archived at *Zagotta, 2021*).

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
