## [Editor Report · eLife Assessment]

This **valuable** study uses fluorescence lifetime imaging and steady-state and time-resolved transition metal ion FRET to characterize conformational transitions in the isolated cyclic nucleotide binding domain of a bacterial CNG channel. The data are **compelling** and support the authors' conclusions. The results advance the understanding of allosteric mechanisms in CNBD channels and have theoretical and practical implications for other studies of protein allostery. A limitation is that only the cytosolic fragments of the channel were studied.

---

## [Referee Report · Reviewer #1 (Public review)]

Summary:

The authors use fluorescence lifetime imaging (FLIM) and tmFRET to resolve resting vs. active conformational heterogeneity and free energy differences driven by cGMP and cAMP in a tetrameric arrangement of CNBDs from a prokaryotic CNG channel.

Strengths:

The data are excellent and provide detailed measures of the probability to adopt resting vs. activated conformations with and without bound ligands.

Weaknesses:

A limitation is that only the cytosolic fragments of the channel were studied.

---

## [Referee Report · Reviewer #2 (Public review)]

The authors investigated the conformational dynamics and energetics of the SthK Clinker/CNBD fragment using both steady-state and time-resolved transition metal ion Förster resonance energy transfer (tmFRET) experiments. To do so, they engineered donor-acceptor pairs at specific sites of the CNBD (C-helix and β-roll) by incorporating a fluorescent noncanonical amino acid donor and metal ion acceptors. In particular, the authors employed two cysteine-reactive metal chelators (TETAC and phenM). This allowed to coordinate three transition metals (Cu2+, Fe2+, and Ru2+) to measure both short (10-20 Å, Cu2+) and long distances (25-50 Å, Fe2+, and Ru2+). By measuring tmFRET with fluorescence lifetimes, the authors determined intramolecular distance distributions in the absence and presence of the full agonist cAMP or the partial agonist cGMP. The probability distributions between conformational states without and with ligands were used to calculate the changes in free energy (ΔG) and differences in free energy change (ΔΔG) in the context of a simple four-state model.

Overall, the work is conducted in a rigorous manner, and it is well-written.

In terms of methodology, this work provides a further support to steady-state and time-resolved tmFRET approaches previously developed by the authors of the present work to probe conformational rearrangements by using a fluorescent noncanonical amino acid donor (Anap) and transition metal ion acceptor (Zagotta et al., eLife 2021; Gordon et al., Biohpysical Journal 2024; Zagotta et al., Biohpysical Journal 2024).

For what concerns Cyclic nucleotide-binding domain (CNBD)-containing ion channels, the literature on this subject is vast and the authors of the present work have significantly contributed to the understanding of the allosteric mechanism governing the ligand-induced activation of CNBD-containing channels, including a detailed description of the energetic changes induced by ligand binding. Particularly relevant are their works based on DEER spectroscopy. In DeBerg et al., JBC 2016, the authors described, at atomic details, the conformational changes induced by different cyclic nucleotides on the HCN CNBD fragment and derived energetics associated with ligand binding to the CNBD (ΔΔG). In Collauto et al., Phys Chem Chem Phys. 2017, they further detailed the ligand-CNBD conformational changes by combining DEER spectroscopy with microfluidic rapid freeze quench to resolve these processes and obtain both equilibrium constants and reaction rates, thus demonstrating that DEER can quantitatively resolve both the thermodynamics and the kinetics of ligand binding and the associated conformational changes.

In the revised manuscript the authors better framed their work in light of the literature by highlighting novelty and limitations, in particular the decision to work with the isolated Clinker/CNBD fragment and not with the full-length protein.

---

## [Referee Report · Reviewer #3 (Public review)]

Summary:

The manuscript by Eggan et al provides insights into conformational transitions in the cyclic nucleotide binding domain of a cyclic nucleotide-gated (CNG) channel. The authors use transition metal FRET (tmFRET) which has been pioneered by this lab and previously led to detailed insights into ion channel conformational changes. Here, the authors not only use steady-state measurements but also time-resolved, fluorescence lifetime measurements to gain detailed insights into conformational transitions within a protein construct that contains the cytosolic C-linker and cyclic nucleotide binding domain (CNBD) of a bacterial CNG channel. The use of time-resolved tmFRET is a clear advancement of this technique and a strength of this manuscript.

In summary, the present work introduces time-resolved tmFRET as a novel tool to study conformational distributions in proteins. This is a clear technological advance. The limitations of the truncated construct used in this study and how they relate to the energetics in full-length CNG channels are discussed. It will be interesting to see in the future how results compare to similar measurements on full-length channels, for example, reconstituted into nanodiscs.

Strengths:

The results capture known differences in promoting the open state between different ligands (cAMP and cGMP) and are consistent across three donor-acceptor FRET pairs. The calculated distance distributions are further in agreement with predicted values based on available structures. The finding that the C-helix is conformationally more mobile in the closed state as compared to the open state quantitatively increases our understanding of conformational changes in these channels.

Weaknesses:

The results describe movements of the C-helix in CNBDs, but detailed energetics as calculated in this study, need to be limited to the truncated protein construct. This is a weakness that cannot be overcome easily as it will require future experiments using the full-length channel.

The data only describe movements of the C-helix. Upon ligand binding, the C-helix moves upwards to coordinate the ligand. Thus, the results are ligand-induced conformational changes (as the title states). Allosteric regulation usually involves remote locations in the protein, which is applicable only in a limited fashion here.

---

## [Author Response]

The following is the authors’ response to the original reviews.

**Reviewer 1:**
Limitations are that only the cytosolic fragments of the channel were studied, and the current manuscript does not do a good job of placing the results in the context of what is already known about CNBDs from other methods that yield similar information.

In the revision, we have now added a paragraph in the discussion that addresses why the cytosolic fragment was used and a paragraph putting our results into the context of previous work on CNBD channels where possible.

(1) Why do the authors not apply their approach to the full-length channel? A discussion of any limitations that make this difficult would be worthwhile.”Full-length ion channel protein expression is more challenging, and it was important to start with a simpler system. This is now stated in the discussion.(2) …nonetheless a comparison of the conformational heterogeneity and energetics obtained from these different approaches would help to place this work in a larger context.

We have now added a paragraph in the discussion putting our work in a larger context and addressing the challenges of comparing our results to previous studies.

(3) Page 5 - 3:1 unlabeled:labeled subunits in mix => 42% of molecules have 3:1 stoichiometry as desired and 21% of molecules have 2:2 stoichiometry!!! (binomial distribution p=0.25, n=4). So 1/3 of molecules with labels have two labeled subunits. This does not seem like it is at all avoiding the problem of intersubunit FRET…

From the experimental perspective, the 3:1 molar ratio stated is certainly a low estimate of the actual subunit ratios given our FSEC data in Figure 2D and the higher expression of the WT protein compared to labeled protein. Furthermore, even without the addition of any WT protein, the calculated contribution of intersubunit FRET is negligible given that the FRET efficiency is heavily dominated by the closest donor-acceptor distances (Figure 4).

(4) Figure 2E - Some monomers appear to still be present in the collected fraction. The authors should discuss any effect this might have on their results.

We now describe in the text that, at the low concentrations (~10nM) used for mass photometry, a second small peak was observed of ~30kDa, which is below the analytical range for this method. This would not affect our results since all tmFRET experiments used higher protein concentrations to ensure tetramerization.

(5) page 4 - "Time-resolved tmFRET, therefore, resolves the structure and relative abundance of multiple conformational states in a protein sample." - structure is not resolved, only a single distance.

We have reworded this sentence.

**Reviewer #2:**
Regarding cyclic nucleotide-binding domain (CNBD)-containing ion channels, I disagree with the authors when they state that "the precise allosteric mechanism governing channel activation upon ligand binding, particularly the energetic changes within domains, remains poorly understood". On the contrary, I would say that the literature on this subject is rather vast and based on a significantly large variety of methodologies…

Despite this vast literature on the energetics of CNBD channels there is no consensus about the energetics and coupling of domains that underlies the allosteric mechanism in any CNBD channel. We have added a separate paragraph in the discussion to clarify our meaning.

In light of the above, I suggest the authors better clarify the contribution/novelty that the present work provides to the state-of-the-art methodology employed (steady-state and time-resolved tmFRET) and of CNBD-containing ion channels……In light of the above, what is the contribution/novelty that the present work provides to the SthK biophysics?

This work is the first use of the time-resolved tmFRET method to obtain intrinsic G (of an apo conformation) and G values for different ligands. It is also the first application of this approach to SthK or, indeed, to any protein other than MBP. This is mentioned in the introduction.

…On the basis of the above-cited work (Evans et al., PNAS, 2020) the authors should clarify why they have decided to work on the isolated Clinker/CNBD fragment and not on the full-length protein…

We chose to start on the C-terminal fragment to provide a technically more tractable system for validating our approach using time-resolved tmFRET before moving to the more challenging full-length membrane protein. This is now addressed in a new paragraph in the discussion.

What is the advantage of using the Clinker/CNBD fragment of a bacterial protein and not one of HCN channels, as already successfully employed by the authors (see above citations)?

We have chosen to perform these studies in SthK rather than a mammalian CNBD channel as SthK presents a useful model system that allows us to later express fulllength channels in bacteria. In addition, the efficiency of noncanonical amino acid incorporation is much higher in bacteria than in mammalian cells.

**Reviewer #3:**

While the use of a truncated construct of SthK is justified, it also comes with certain limitations…

We agree that the truncated channel comes with limitations, but we still think that there is relevant energetic information from studies of the isolated CNBD. This is now addressed in the discussion.

I recommend the authors carefully assess their statements on allostery. …The authors also should consider discussing the discrepancies between their truncated construct and full-length channels in more detail.

We added a paragraph in the introduction that now puts the conformational change of the CNBD in the context of the allosteric mechanism of the full-length channel. We also added a paragraph discussing in more detail the relationship between the energetics of the C-terminal fragment and the full-length channel.

Regarding the in silico predictions, it is unclear to me why the authors chose the closed state of SthK Y26F and the 'open' state of the isolated C-linker CNBD construct…

The active cAMP bound structure (4d7t) was a high resolution X-ray crystallography structure chosen as the only model with a fully resolved C-helix. The resting state structure (7rsh) was selected as a the only resting state to resolve the acceptor residue studied here (V417).

Previously it has been shown that SthK (and CNG) goes through multiple states during gating. This may be discussed in more detail, especially when it comes to the simplified four-state model…

As stated above, we added paragraphs to the introduction and discussion placing the conformational change of the CNBD in the context of the full-length channel.

It would be interesting to see how the conformational distribution of the C-helix position integrates with available structural data on SthK. In general, putting the results more into the context of what is known for SthK and CNG channels, could increase the impact.

We now discuss the relationship between existing structures and energetics in the introduction.

This may be semantics, but when working with a truncated construct that is missing the transmembrane domains using 'open' and 'closed' state is questionable. I recommend the authors consider a different nomenclature.

We refer to the conformational states of the CNBD as ‘resting’ and ‘active’ and used ‘closed’ and ‘open’ only for the conformational states of the pore.